# Multiomic analysis of human kidney disease identifies a tractable inflammatory and pro-fibrotic tubular cell phenotype

Maximilian Reck [1,14], David P. Baird [2,14], Stefan Veizades [1,3], Callum Sutherland[1], Rachel M. B. Bell[1], Heeyoun Hur[1], Carolynn Cairns[1], Piotr P. Janas [2], Ross Campbell [2], Andy Nam [4], Wei Yang[4], Nathan Schurman [4], Claire Williams [4], Eoin O'Sullivan [5,6], Meryam Beniazza[7], Andrea Corsinotti[7], Christopher Bellamy [2], Jeremy Hughes[2], Alexander Laird [8,9], Laura Denby[1], Tamir Chandra [10,11,12,13], David A. Ferenbach [2] & Bryan R. Conway [1] ✉

Maladaptive proximal tubular (PT) epithelial cells have been implicated in progression of chronic kidney disease (CKD), however the complexity of epithelial cell states within the fibrotic niche remains incompletely understood. Hence, we integrated snRNA and ATAC-seq with high-plex single-cell molecular imaging to generate a spatially-revolved multiomic atlas of human kidney disease. We demonstrate that in injured kidneys, a subset of *HAVCR1*+*VCAM1*+ PT cells acquired an inflammatory phenotype, upregulating genes encoding chemokines, pro-fibrotic and senescence-associated proteins and adhesion molecules including *ICAM1*. Spatial transcriptomic and multiplex-immunofluorescence determined that specifically these VCAM1+ICAM1+ inflammatory PT cells localised to the fibrotic niche. Ligand-receptor analysis highlighted paracrine signaling from inflammatory PT cells mediating leucocyte recruitment and myofibroblast activation. Loss of HNF4α and activation of NF-κB and AP-1 transcription factors epigenetically imprinted the inflammatory phenotype. Targeting inflammatory tubular cells by administering an AP-1 inhibitor or senolytic agent ameliorated inflammation and fibrosis in murine models of kidney injury, hence these cells may be a tractable target in CKD.

Human kidneys comprise multiple different cell types in a complex anatomical arrangement that is optimal for their key functions including removal of waste, regulation of fluid and electrolyte homeostasis and production of hormones. In chronic kidney disease (CKD), this arrangement is disrupted with de-differentiation and atrophy of epithelial cells, recruitment of leukocytes and activation of myofibroblasts[1,2]. An improved understanding of the spatial arrangement of these cells and their

[1]Centre for Cardiovascular Science, University of Edinburgh, Edinburgh, UK. [2]Centre for Inflammation Research, University of Edinburgh, Edinburgh, UK. [3]Stanford Cardiovascular Institute, Stanford University, Stanford, CA, USA. [4]Bruker Spatial Biology, Seattle, WA, USA. [5]Institute of Molecular Bioscience, University of Queensland, Brisbane, QLD, Australia. [6]Kidney Health Service, Metro North Hospital and Health Service, Brisbane, QLD, Australia. [7]Institute for Regeneration and Repair, University of Edinburgh, Edinburgh, UK. [8]Institute of Genetics and Cancer, University of Edinburgh, Edinburgh, UK. [9]Department of Urology, NHS Lothian, Edinburgh, UK. [10]MRC Human Genetics Unit, University of Edinburgh, Edinburgh, UK. [11]Robert and Arlene Kogod Center on Aging, Mayo Clinic, Rochester, MN, USA. [12]Department of Biochemistry and Molecular Biology, Mayo Clinic, Rochester, MN, USA. [13]Department of Quantitative Health Sciences, Mayo Clinic, Rochester, MN, USA. [14]These authors contributed equally: Maximilian Reck, David P. Baird. ✉e-mail: bryan.conway@ed.ac.uk

intercellular signalling pathways could lead to development of new therapies.

Recent advances in single-cell technologies including single nuclear RNA-sequencing (snRNA-seq) and assay for transposase-accessible chromatin sequencing (ATAC-seq) have facilitated the development of high-resolution datasets of the human and murine kidney in health and disease[3–8]. Studies in humans have characterized an adaptive proximal tubular (PT) cell phenotype, characterized by expression of *VCAM1*, that is present in the normal kidney but expanded in acute kidney injury (AKI) and CKD[4,8–11]. In murine studies, *Havcr1+Vcam1+* maladaptive or 'failed repair' PT cells express pro-inflammatory, pro-fibrotic and senescence-associated genes and persist after ischaemic renal injury[12,13]. However, the relevance of the failed repair phenotype to human kidney disease and particularly its spatial context is currently unknown.

To this end, we performed simultaneous snRNA-seq, ATAC-seq and high-plex single-cell molecular imaging of non-tumour bearing tissue from tumour nephrectomy specimens from patients with otherwise healthy kidneys and those in which the tumour obstructed the ureter leading to inflammatory fibrosis. Specifically in obstructed kidneys, a subset of the *VCAM1*⁺ adaptive PT cells adopted an inflammatory phenotype, up-regulating adhesion molecules such as *ICAM1*, chemokines and genes consistent with activation of a senescence program. High-plex single-cell molecular imaging localized this inflammatory phenotype to the fibrotic niche in obstructed kidneys and in biopsies from patients with IgA nephropathy. Multiplex immunofluorescence confirmed that specifically the ICAM-1⁺ inflammatory subset co-localises with immune cells and myofibroblasts in the fibrotic niche. Integration of gene expression and ATAC data implicated HNF4-α in maintaining proximal tubular cell health and AP-1 signalling in promoting the inflammatory tubular phenotype. Administration of a small molecule inhibitor of AP-1 or the senolytic agent ABT-263 reduced expression of genes consistent with an inflammatory tubular cell phenotype and ameliorated inflammation and fibrosis in murine models of transition from acute kidney injury (AKI) to CKD.

## Results

### Generation of an atlas of healthy and obstructed kidneys

To characterize the cellular and molecular landscape in the healthy and injured human kidney, we performed simultaneous snRNA-seq and ATAC-seq and high-plex single-cell molecular imaging on tissue from the non-tumorous pole of nephrectomies from 5 patients with urothelial carcinoma causing unilateral ureteric obstruction (UUO) as a paradigm of tubular injury, with 7 patients who had renal cell carcinoma without urinary obstruction constituting our controls (Fig. 1a). The clinical and pathological characteristics of the patients are given in Supplementary Data 1, 2 and Supplementary Fig. S1. All controls had normal renal function (eGFR >75 ml/min/m²) at baseline and exhibited a decline in kidney function post-operatively. Conversely, in 4 of the 5 obstructed patients, the renal function remained stable post-operatively indicating the resected kidney was non-functional due to complete ureteric obstruction (Supplementary Fig. S1a), while in the remaining case there was a decline in kidney function post-operatively suggesting incomplete ureteric obstruction. In both the controls and obstructed specimens there was mild glomerulosclerosis, commensurate with the cohort age (mean 71 years); however, the obstructed samples exhibited significantly greater tubulointerstitial fibrosis and atrophy (Supplementary Data 1, Supplementary Fig. S1b).

To mitigate against batch effects, we isolated nuclei from snap-frozen wedge biopsies from each patient and constructed 6 pools, each comprising nuclei from 3 to 5 patients, with most patients represented in more than one pool (Supplementary Fig. S2a, b, Supplementary Data 2). We performed combined snRNA-seq and ATAC-seq using the Chromium Multiome Assay (Fig. 1a, Supplementary Fig. S2b–g) and pooled cells were then assigned to each donor by computational SNP-based deconvolution (Supplementary Fig. S2a). By leveraging SNP variants detected in sequencing reads, we assigned cells to donors for both the gene expression and ATAC libraries. Comparison of inferred donors between libraries showed a high (98.68%) donor agreement across gene expression and ATAC modalities (Supplementary Fig. S2h). Predicted donors were matched to original samples by leveraging the pooling pattern created during library construction, with samples pooled across multiple libraries showing a low genotype distance (Supplementary Fig. S2i, k). To confirm the correct assignment of cells to donors, SNP microarray genotyping was performed on surplus tissue, with paired samples showing a low genotype distance, hence validating the accuracy of our approach (Supplementary Fig. S2j). The pooling approach facilitated detection of nuclei doublets, identified by the presence of SNPs characteristic of different donors. Indeed, not all doublets would have been excluded by standard methods such as applying a maximum threshold to the gene or peak count (Supplementary Fig. S2g).

Following quality control (Supplementary Fig. S3), we analysed joint transcriptomes and chromatin state of 46,957 nuclei, identifying 59 discrete clusters (Fig. 1b, Supplementary Data 3) which were annotated using the differentially expressed genes and linked differentially accessible chromatin regions in each cluster (Supplementary Figs. S4, S5, Supplementary Data 4). To assess how our clustering compared with existing datasets, we projected snRNA-seq data from the Kidney Precision Medicine Project (KPMP) onto our dataset, confirming high correlation between cells assigned to comparable clusters across both datasets (Supplementary Fig. S6). Within our tubular cell clusters, cells from obstructed kidneys segregated from controls (Fig. 1c) and were characterized by a core injury gene signature including *PROM1, DCDC2, SPP1, ITGB6 and ITGB8* (Fig. 1d) consistent with that of cells annotated as 'adaptive' in the KPMP atlas[4]. Hence following ureteric obstruction, injured tubules adopt an injury response that is conserved across the ischaemic, diabetic and hypertensive nephropathies assessed in the KPMP atlas. The obstructed kidneys contained proportionally fewer healthy tubular cells than unobstructed controls and were enriched for this injured tubular cell phenotype (Fig. 1e). In addition, there were fewer peritubular capillary endothelial cells in obstructed kidneys consistent with rarefaction of the peri-tubular capillary network (Fig. 1e). Conversely, in obstructed kidneys there was a higher proportion of myeloid cells, T-cells and B-cells and the CD56^Bright subset of natural killer cells, in keeping with prior studies implicating this subset in progression of CKD[14]. In addition, there was evidence of a phenotypic switch in stromal cells with fewer resident fibroblasts and macrophages and an increase in myofibroblasts and activated macrophages (Fig. 1e). A limitation of using snRNA-seq to quantify cell states is loss of spatial context and that some cell types may be differentially liberated or damaged during nuclear isolation. However, by employing high-plex single-cell molecular imaging using ~6,000 transcripts on the CosMx platform we were able to spatially annotate the key cell populations and confirm that there was a greater proportion of injured tubular cells, immune cells and interstitial cells in obstructed kidneys compared with unobstructed controls (Fig. 1f, Supplementary Figs. S7, 8).

### Identification of an inflammatory subset of tubular cells specific to injured kidneys

To explore the phenotypes of the tubular cell clusters in more detail, we segregated and re-clustered each tubular cell phenotype. Proximal tubular (PT) cells comprised healthy cells, corresponding to the S1-3 segments and injured cells which expressed *HAVCR1* and *VCAM1* (Fig. 2a–c, Supplementary Data 5) and were enriched in UUO samples. When projected onto the KPMP snRNA-seq atlas[4] the *HAVCR1*⁺ *VCAM1*⁺ injured PT cells corresponded to the cluster denoted as 'adaptive' PT cells (Fig. 2d, e, Supplementary Fig. S9a, b). Additionally, we

detected a subset of the injured PT cells which was almost exclusively detected in obstructed kidneys and expressed chemokines (e.g. *CCL2*, *CXCL1*), pro-fibrotic factors (e.g. *TGM2*) and adhesion molecules (e.g. *ICAM1*), which we denoted as inflammatory PT cells (Fig. 2a-c, Supplementary Data 5). We further interrogated the adaptive PT cells in the KPMP Atlas, confirming the presence of an *ICAM1*+ subset that expresses the same inflammatory gene signature (Fig. 2d, e, Supplementary Fig. S9c, d), suggesting that this PT cell subset is found in other kidney diseases. In patients with CKD recruited to the KPMP atlas, the proportion of PT cells that aligned to our inflammatory phenotype correlated closely with the proportion of activated

macrophages and myofibroblasts (Supplementary Fig. S9e), whereas the correlation between total adaptive cells and macrophages/myofibroblasts was non-significant. Hence, the *ICAM1*+ inflammatory PT cells represent a subset of adaptive tubular cells, which we wished to characterise in more detail given their inflammatory gene signature and correlation with immune cells and myofibroblasts.

Trajectory analysis demonstrated that following injury, PT cells down-regulated genes encoding key tubular cell functions such as ion transport and metabolism (Fig. 2f–h, Supplementary Data 6) and expressed the core tubular injury signature (Fig. 2c, g). RNA velocity suggested that cells in the initial stages of the injury trajectory may

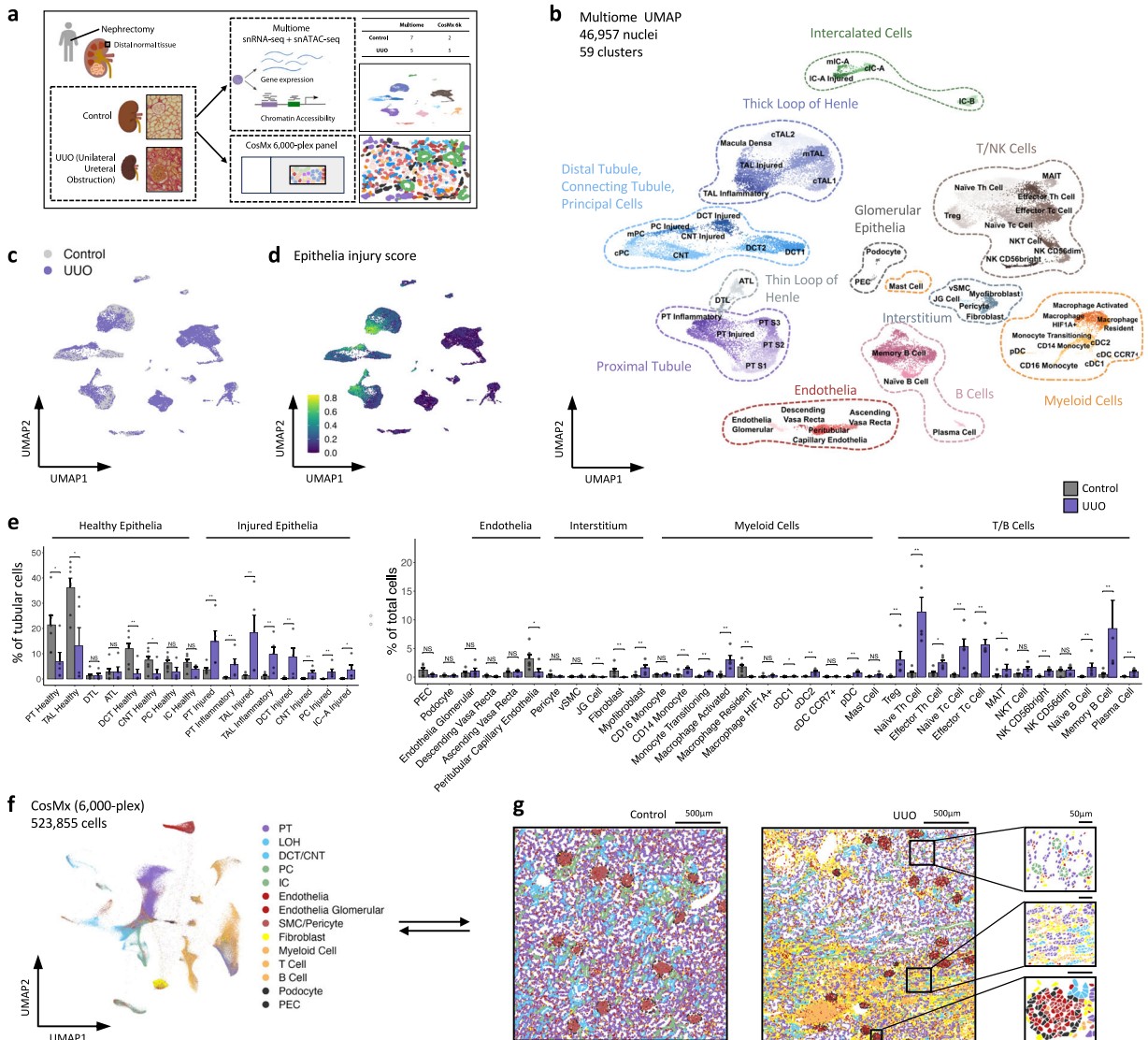

**Fig. 1 | A multiomic single-nuclei atlas of kidney injury. a** Overview of the single nuclear multiome (paired Assay for Transposase Accessible Chromatin for sequencing (snATAC-seq) and snRNA-seq) as well as sub cellular-resolution spatial transcriptomics (CosMx SMI 6000 transcript panel) workflows. For the multiome assay nuclei were isolated from snap-frozen human nephrectomy specimens from control (n = 7) or unilateral ureteric obstructed (UUO, n = 5) kidneys and processed in pools of three to five samples per library using the 10x Genomics Multiome Assay. Adjacent FFPE tissue was used for spatial transcriptomics analysis for a subset of samples (UUO, n = 5; Control, n = 2). **b** Weighted-nearest neighbour UMAP projection of joint transcriptome and open chromatin state of 46,957 nuclei with detailed cluster annotations. PEC parietal epithelial cell, PT proximal tubule, DTL descending thin limb, ATL ascending thin limb, TAL thick ascending limb, DCT distal convoluted tubule, CNT connecting tubule, PC principal cell, IC intercalated

cell, vSMC vascular smooth muscle cell, JG cell juxtaglomerular cell, pDC plasmacytoid dendritic cell, cDC classical dendritic cell, Th T helper cell, Tc cytotoxic T cell. **c** UMAP projection annotating cells to control or UUO kidneys. **d** Gene expression score of epithelial injury genes universally expressed in injured nephron segments (*PROM1, DCDC2, SPP1, ITGB6, ITGB8*). **e** Percentage of nuclei assigned to each cell type as a proportion of all epithelial cells (for epithelial clusters, left) and of total cells (for non-epithelial clusters, right) in control (n = 7) and obstructed (n = 5) kidneys. Plots show means ± standard error of the mean (SEM). Wilcoxon rank-sum test. *p < 0.05; **p < 0.01. **f** UMAP plots of the CosMx 6000-plex dataset showing cells coloured by their broad annotations. **g** Representative figures of cells from control and UUO kidneys plotted in 2D space. Cells are coloured according to the broad cell types (Fig. 1f). Source data are provided as a Source Data file.

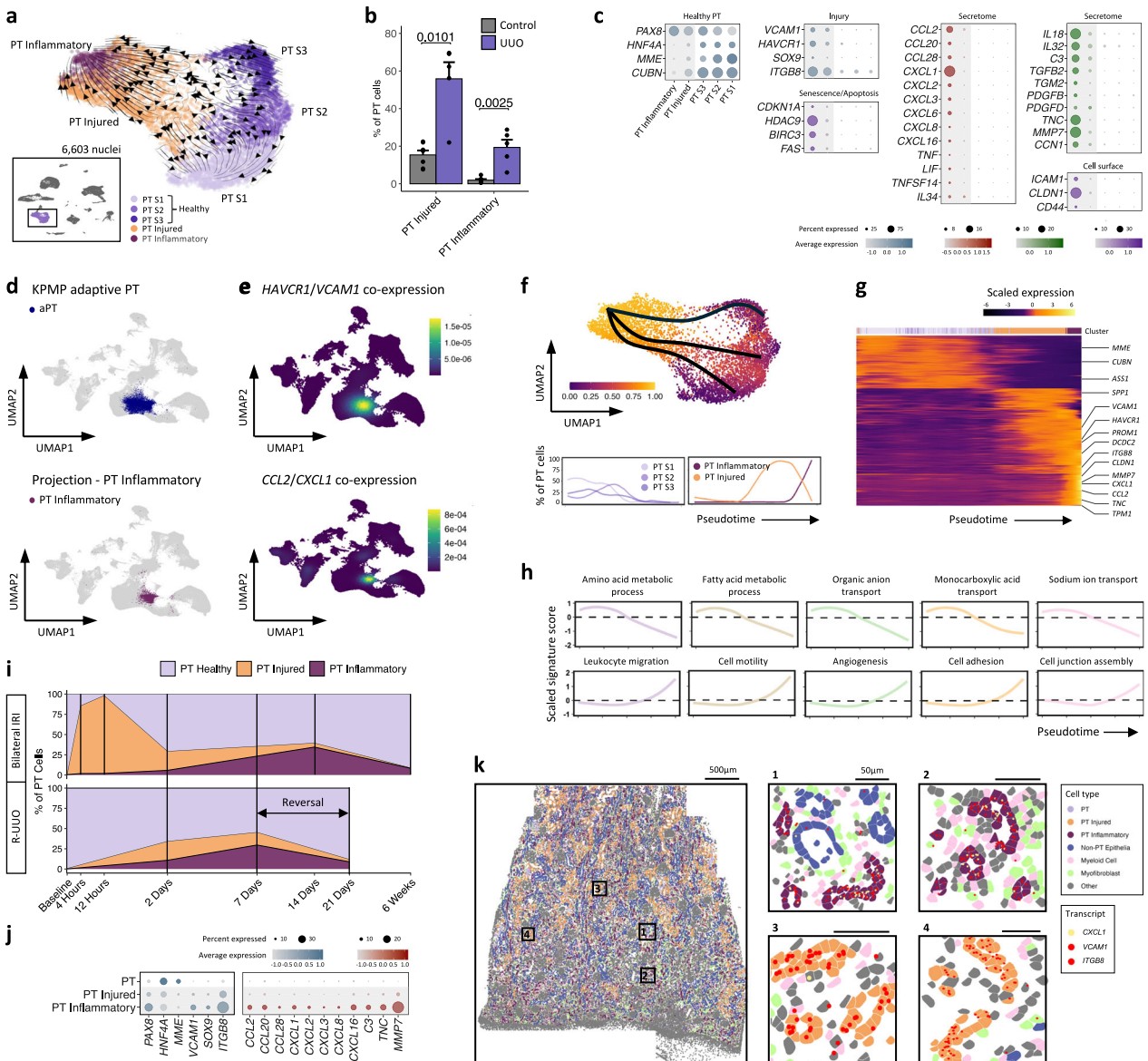

**Fig. 2 | Injured PT cells can adopt a pro-inflammatory, pro-fibrotic phenotype.**
**a** UMAP of proximal tubule (PT) cells coloured by cell phenotype and projected RNA velocities (arrows). **b** Barplot showing mean (±SEM) percentage of injured and inflammatory PT cells in control ($n = 7$) and obstructed (UUO, $n = 5$) kidneys as a proportion of total PT cells. Wilcoxon rank-sum test. **c** Dot plot of selected differentially expressed genes in healthy, injured and inflammatory PT cells. Dot colours show the averaged gene expression values (log scale); size indicates proportion of cells expressing each gene. **d** Identification of inflammatory PT cells in the Kidney Precision Medicine Project (KPMP) snRNA-seq dataset[4]. Top panel: original adaptive PT (aPT) annotations; bottom panel: KPMP cells adopting an inflammatory PT cell phenotype. **e** Density plot highlighting cells co-expressing injury (*HAVCR1, VCAM1*) and inflammatory (*CCL2, CXCL1*) cell specific transcripts. **f** Top panel: UMAP (as in Fig. 2a) with cells ordered in pseudotime on a trajectory from healthy PT segments to the inflammatory cell state. Bottom panel: percentage of cells in each subcluster as a proportion of total PT cells at that trajectory point. **g** Heatmap of

scaled gene expression along the pseudotime trajectory. **h** Signature scores of GO terms enriched in healthy (PT S1-S3) or injured and inflammatory PT cells.
**i** Proportion of total PT cells adopting a healthy, injured and inflammatory phenotype at different timepoints in single-cell/nuclei datasets in murine models of ischaemia-reperfusion injury (IRI, top[9]) and reversible unilateral ureteric obstruction (R-UUO, bottom[15]). **j** Genes differentially expressed in inflammatory PT cells in the CosMx (6000-plex panel) dataset. The dot colours show the averaged gene expression values (log scale) and size indicates proportion of cells expressing each gene. **k** Representative image of obstructed tissue section analysed by CosMx, with dots representing cells in their 2D coordinates. PT cells are coloured according to cell state, with selected other cell types highlighted. Plots on the right show high-resolution cell segmentation boundaries in areas enriched with inflammatory (top) or injured PT cells (bottom). Dots represent individual injury-associated (red) and inflammatory-associated (yellow) transcripts. Source data are provided as a Source Data file.

retain the potential to revert towards a healthy phenotype, while cells in a more advanced injury state transitioned exclusively towards an inflammatory phenotype (Fig. 2a). To determine whether the trajectory analysis correlated with cell dynamics in pre-clinical models, we compared the human PT cell phenotypes with scRNA-seq datasets from two murine models – ischaemic reperfusion injury (IRI)[9] and reversible unilateral ureteric obstruction (R-UUO)[15]. We observed

conserved injured and inflammatory PT cell phenotypes across species, with the inflammatory PT subset mapping to cells analogous to those previously designated as maladaptive or 'failed repair'[9] (Supplementary Fig. S10). Following injury, there was a rapid increase in the proportion of injured tubular cells, with the inflammatory cell phenotype appearing later (Fig. 2i, Supplementary Fig. S10c). In both models, the proportion of injured PT cells fell progressively during the

recovery period while the proportion of healthy PT cells increased, suggesting reversion of injured to healthy tubules. In contrast, the inflammatory PT cells persisted at a higher level for longer after injury.

Taken together, the human and murine data indicate a 'point-of-no-return' beyond which injured PT cells adopt an inflammatory phenotype that is incapable of reverting towards a healthy phenotype and persists despite cessation of injury. These features are characteristic of senescent renal epithelia[16] and we noted that the inflammatory PT cells also expressed senescence-associated genes (Fig. 2c) including *CDKN1A* and *HDAC9*[17], which has recently been shown to mediate renal fibrosis[18]. To assess this further, we induced senescence by irradiating renal proximal tubular epithelial cells (RPTECs) and performed bulk RNA-seq to identify genes that were > 2-fold up-regulated in senescent compared with healthy RPTECs (Supplementary Fig. S11a–d). The signature score of these genes was markedly higher in the inflammatory PT cells (Supplementary Fig. S11e, f), consistent with the transcriptional activation of a senescence program.

We sought to determine whether similar inflammatory phenotypes were observed for tubular epithelial cells in other nephron segments in obstructed kidneys. We observed a subset of injured cells in the thick ascending limb of the Loop of Henle that expressed multiple chemokine, adhesion and pro-fibrotic genes comparable to the inflammatory PT cells (Supplementary Fig. S12, Supplementary Data 5, 7). Other injured tubular cell types, including connecting tubules and principal cells expressed a modest number of inflammatory transcripts, though this was less marked in distal convoluted tubular cells (Supplementary Fig. S12, Supplementary Data 8).

Lastly, we investigated if inflammatory PT cell states can be identified using spatial molecular imaging. We identified discrete injured and inflammatory PT cell phenotypes, which exhibited similar differentially expressed genes as in the snRNA/ATAC-seq analysis (Fig. 2j, Supplementary Data 9–11). Visualisation of cell states in 2D space revealed that tubules adopt either injured or inflammatory phenotypes without much intermixing of cell states within individual tubules (Fig. 2k).

## Inflammatory tubular cells co-localise with monocytes and myofibroblasts in the fibrotic niche

Given inflammatory cell states can be identified in discrete tubules, we next sought to determine the spatial association of PT cell states with immune cells and myofibroblasts. The inflammatory PT cells, but not injured PT cells, co-localised with myeloid cells and myofibroblasts (Fig. 3a, Supplementary Fig. S13). To determine the degree of co-localisation of individual cell types, we quantified the pairwise co-localisation within a 25 μm radius compared to that expected in a random topographical distribution (Fig. 3b). There was enrichment of monocytes, transitioning monocyte-macrophages, dendritic cells and myofibroblasts adjacent to the inflammatory but not injured PT cells (Fig. 3c, Supplementary Data 12). To confirm this finding, we assessed whether transcripts associated with myeloid cells and myofibroblasts were enriched in spatial proximity to inflammatory PT cell centroids. Indeed, transcripts up-regulated by inflammatory PT cells were highly-enriched within cell boundaries (within a 0–5 μm radius of cell centroids) (Fig. 3d), whereas archetypal transcripts characteristic of myofibroblasts, monocytes, macrophages and dendritic cells were enriched immediately outside inflammatory PT cell boundaries (> 5 μm of cell centroids) (Fig. 3d).

To develop a panel of antibodies to localise the injured and inflammatory PT cell phenotypes, we leveraged the differentially expressed genes from our snRNA-seq data (Fig. 2c), designating injured and inflammatory PT cells as VCAM1⁺ICAM1⁻ and VCAM1⁺ICAM1⁺, respectively. VCAM1⁺ICAM1⁺ inflammatory PT cells localized to the fibrotic niche adjacent to CD68⁺ macrophages and fibroblast-activated protein (FAP)⁺ myofibroblasts, whereas the VCAM1⁺ICAM1⁻ injured PT cells were more diffusely scattered across

the tissue (Fig. 3e–l), replicating the findings from the CosMx analysis. Intriguingly, ICAM1 was expressed specifically on the luminal membrane of inflammatory PT cells (Fig. 3f), where it may mediate tethering of myeloid cells in the urinary space (Fig. 3h) by binding to receptors including CD11b (*ITGAM*) and CD11c (*ITGAX*). We developed a method to quantify the spatial relationships between cell types (Fig. 3i–k), which determined that both myofibroblasts and macrophages were preferentially located adjacent to VCAM1⁺ICAM1⁺ inflammatory tubules (Fig. 3l).

## Inflammatory PT cell states are found in different CKD aetiologies

To determine whether our findings were applicable to other kidney diseases, we performed high-plex single-cell molecular imaging using a panel of ~1000 probes on renal biopsy tissue from patients with minimal change disease (*n* = 3) or IgA nephropathy (*n* = 6) and nephrectomy specimens from patients with inflammatory fibrosis due to recurrent pyelonephritis (*n* = 4, Fig. 4a, b, Supplementary Data 14–17, Supplementary Data 13). We again identified injured and inflammatory PT cell phenotypes and we determined that it was specifically tubules adopting an inflamed, but not injured phenotype, that were surrounded by immune cells and myofibroblasts in the fibrotic niche (Fig. 4c–h, Supplementary Data 14–17). Importantly, the proportion of PT cells that adopted an inflammatory phenotype in each biopsy correlated with the proportion of myofibroblasts, monocytes and dendritic cells (Fig. 4i) and the severity of fibrosis (Supplementary Fig. S17) and was inversely proportional to renal function (Fig. 4j, Supplementary Data 18). Similar findings were observed for the inflammatory LOH cells (Supplementary Fig. S17).

Hence, our data implicate inflammatory PT cells in recruitment of immune cells and activation of myofibroblasts to promote inflammation and interstitial fibrosis in patients with CKD of diverse aetiology.

## Inter-cellular signalling pathways between inflammatory tubules, leucocytes and myofibroblasts

We next sought to identify signalling pathways by which inflammatory tubular cells communicate with adjacent cell types. Ligand-receptor analysis of the snRNA-seq data demonstrated that inflammatory PT cells expressed a broad range of chemokines, adhesion molecules and complement to attract and activate diverse immune cell types (Fig. 5a, Supplementary Data 19). To determine the primary sources of chemokines locally promoting fibrosis we divided the tissue into niches, with the fibrotic niche being enriched with inflammatory PT cells, myeloid cells and myofibroblasts (Fig. 5b–d). Inflammatory PT cells were the principal source of the chemokine signature in the fibrotic niche, (Figs. 2c and 5e) and were located adjacent to myeloid cells thereby facilitating cell-cell cross-talk[19] to promote leucocyte recruitment and activation (Fig. 5f, g).

In addition, inflammatory PT cells expressed multiple ligands for fibroblast PDGF, TGF and FGF receptors (Fig. 5h), which have been implicated in fibroblast proliferation and activation to myofibroblasts[20–22]. Indeed, in our spatial analysis, multiple PDGF transcripts were identified in inflammatory tubules adjacent to myofibroblasts expressing *PDGFRA* (Fig. 5i, j). In addition, *FGFR1* was detected in activated macrophages, which is of interest as FGF receptor inhibition ameliorates LPS-induced macrophage activation, inflammation and fibrosis in the kidney[23,24].

Lastly, we investigated how inflammatory PT cells may be influenced by adjacent cells. Receptors for FAS and TWEAK were up-regulated in inflammatory PT cells (Fig. 5k), while the ligands were expressed by NK cells (FAS) and monocytes (TWEAK), with both these cell types being implicated in kidney disease[19,25]. While these are classically considered pro-apoptotic factors, they can also promote inflammation via NF-κβ signalling[26,27] and hence enable bi-directional

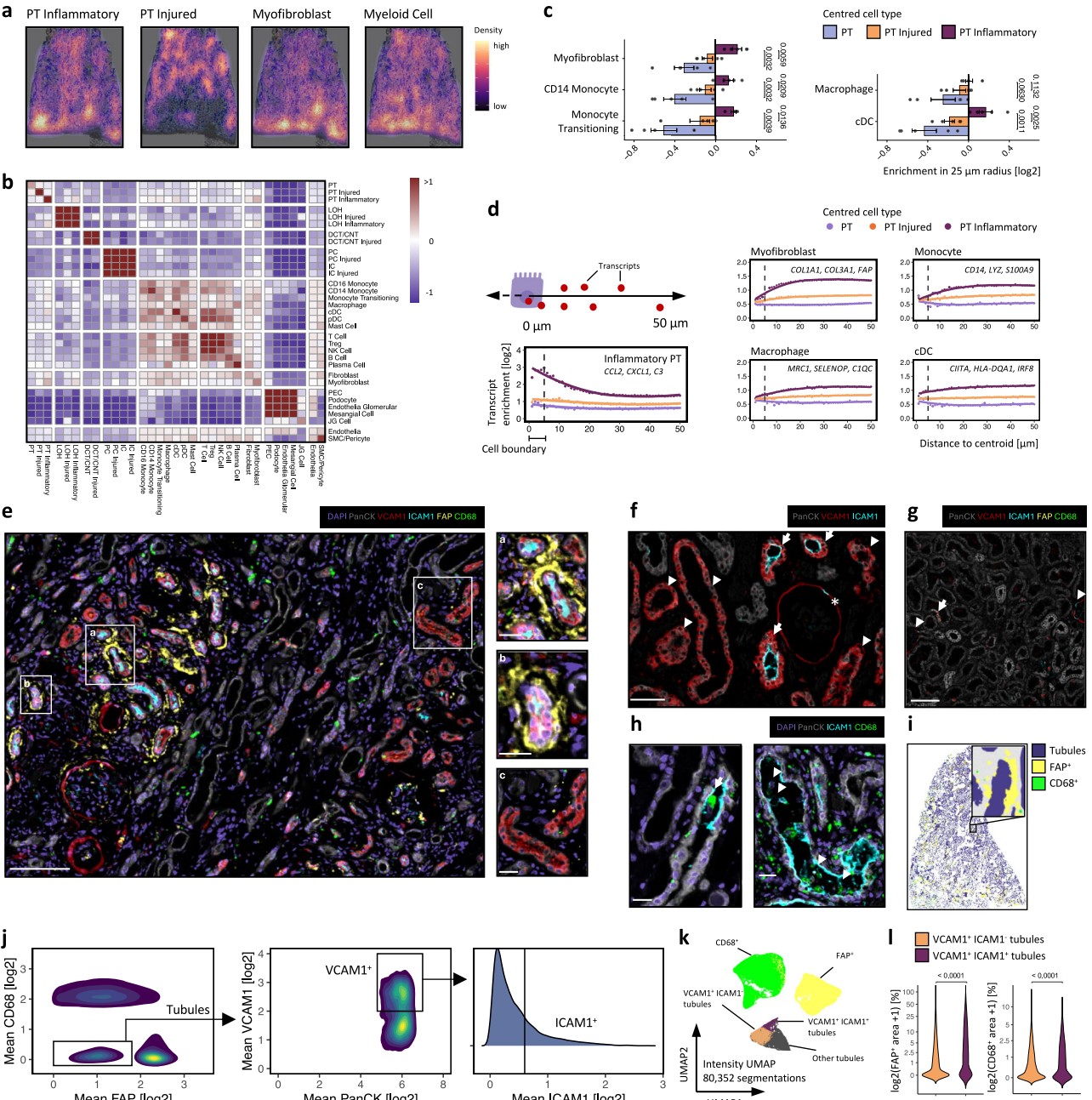

**Fig. 3 | Inflammatory PT cell states are associated with the fibrotic niche in human kidney disease. a** Density maps of cell type abundance overlaid on the same tissue section as in Fig. 2k. **b** Cell neighbourhoods in UUO samples ($n = 5$). Heatmap shows the mean enrichment (log2 scale, mean weighted by cell type abundance) of each cell type within a 25 μm radius of the reference cell type. Values greater or less than zero indicate enrichment or depletion of the cell type. cDC, classical dendritic cell. **c** Bar plots showing the enrichment ratio of myofibroblasts and myeloid cell types in proximity to inflammatory cell states in UUO samples ($n = 5$). Plots show means ± SEM. Paired two-sample Student's $t$ test. **d** Mean enrichment ratio of indicated transcripts adjacent to healthy, injured and inflammatory PT cell states relative to randomly sampled cells. x-axis: distance from centroids of all cells of that group, with transcripts quantified in 1 μm intervals. **e** Representative image of 5-plex immunofluorescence of obstructed nephrectomy tissue (from $n = 4$ samples), with insets showing higher magnification images. Scale bars: large image, 100 μm; small images, 20 μm. **f** 3-plex immunofluorescence

demonstrating VCAM1+/ICAM1+ (arrow) and VCAM1+/ICAM1- tubules (arrowhead) and the parietal epithelium of a glomerulus (*). Scale bar, 50 μm. **g** 5-plex immunofluorescence of healthy kidneys (from $n = 2$ samples), highlighting VCAM1+ tubular cell (arrow) and VCAM1+ parietal epithelial cells (arrowhead). Scale bar, 100 μm. **h** Immunofluorescence of obstructed kidney with CD68+ cells (arrows, left) in the lumen of ICAM1+ tubules. Arrowheads highlight CD68+ cells filopodia anchoring to ICAM1+ tubular cells (right). Scale bar, 20 μm. **i** Representative image of tubule (PanCK+), myofibroblast (FAP+) and macrophage (CD68+) segmentation. **j** Tubules were identified as FAP-/CD68-/PanCK+ and the proportion staining for VCAM1 and ICAM was determined. **k** UMAP of segmentation objects according to staining intensities of PanCK, VCAM1, ICAM1, FAP and CD68. **l** Enrichment of FAP and CD68 in proximity to VCAM1+/ICAM1- or VCAM1+/ICAM1+ tubules. Violin plots of the percentage of the area covered (log2 scale) by FAP+ or CD68+ segmentations objects within a 25 μm radius of 6,304 tubule boundaries. Wilcoxon rank-sum test. Source data are provided as a Source Data file.

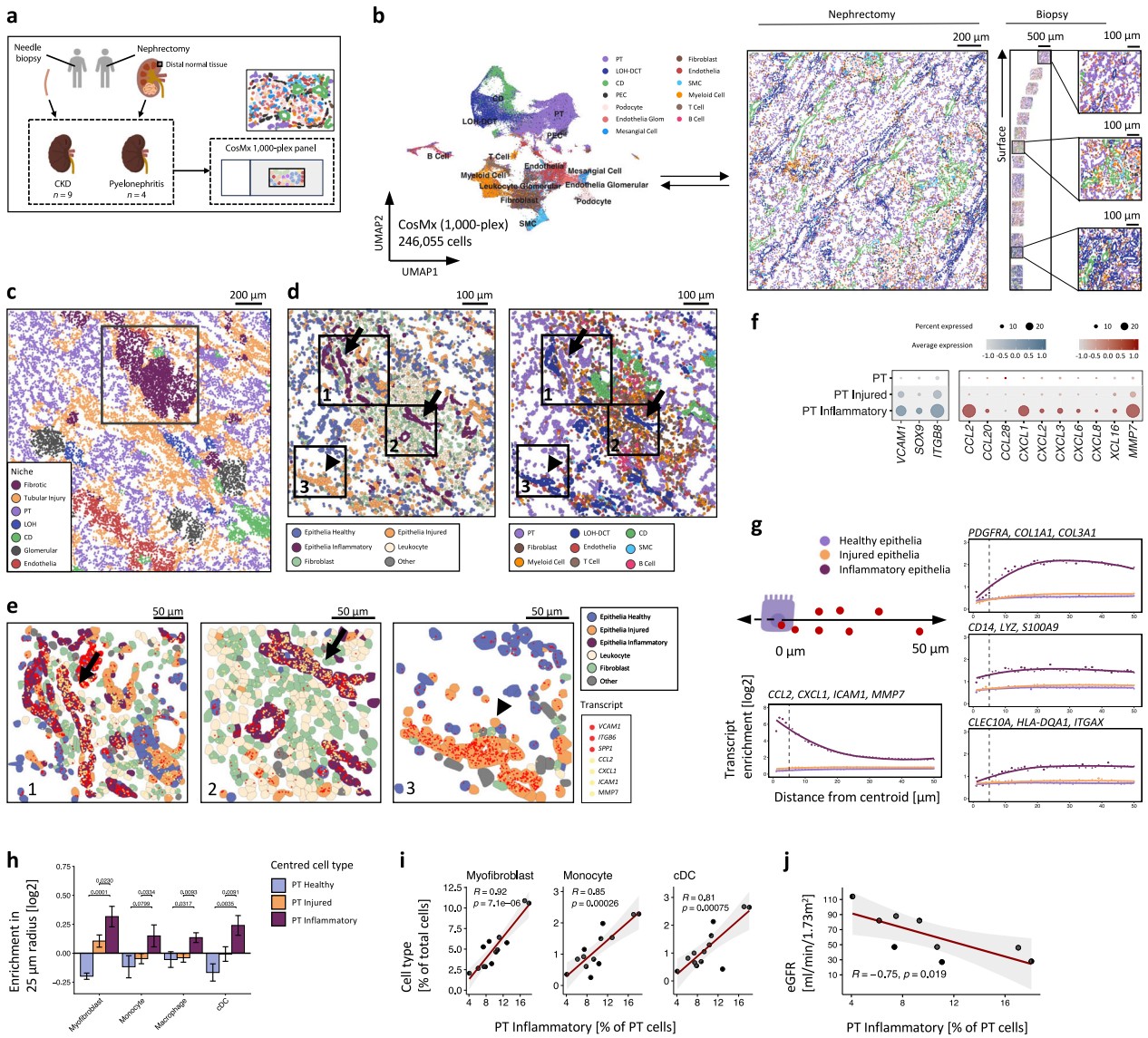

**Fig. 4 | Inflammatory cell states in the kidney across disease aetiologies.**
**a** Workflow to spatially quantify 1,000 transcript species on the CosMx platform in nephrectomy samples from patients with chronic pyelonephritis ($n = 4$) and renal biopsies from patients with minimal change disease ($n = 3$) or IgA nephropathy ($n = 6$). **b** Left: UMAP visualisation of 246,055 cells across all samples coloured by cell annotations. PEC parietal epithelial cell, PT proximal tubular cell, LOH-DCT Loop of Henle and distal convoluted tubule, CD Collecting Duct, SMC Smooth muscle cell. Right: Representative images of nephrectomy and biopsy samples, showing cells coloured by annotation key in UMAP. **c** Representative spatial plot of pyelonephritic kidney. Dots represent individual cells coloured by their niche cluster. **d** Higher resolution of the rectangular inset in (**c**). highlighting inflammatory (arrows) and injured (arrowhead) tubular epithelia. Left: Cells coloured according to activation state and broad cell lineage. Right: Cells coloured according to granular annotation. **e** Higher resolution of areas indicated in (**d**) highlighting inflammatory (arrow) and injured (arrowhead) epithelia. Dots show

the location of injury-associated (red) or inflammatory-associated (yellow) transcripts with cells coloured by activation state and broad lineage. **f** Dotplot of expression of injury and inflammatory markers identified by snRNA-seq (Fig. 2c) in the CosMx (1000-plex) dataset. Dot colours show the averaged gene expression (log scale) and size indicates proportion of cells expressing the gene. **g** Mean enrichment ratio of transcripts in healthy, injured and inflammatory PT cell states relative to randomly sampled cells. x-axis: distance from centroids of all cells of the respective group, with transcripts quantified in $1\,\mu m$ intervals. **h** Bar plots showing enrichment of myofibroblasts and myeloid cell types in proximity to PT cell states. Plots show means ± SEM. Paired two-sample Student's $t$ test. **i** Correlation between percentage of inflammatory PT cells and the proportions of myofibroblasts, monocytes and classical dendritic cells (cDC) in biopsy samples. Graph shows linear regression slope with Pearson co-efficient. **j** Correlation between percentage of inflammatory PT cells and estimated glomerular filtration rate (eGFR) in biopsy samples. Linear regression slope with Pearson co-efficient.

cross talk between leucocytes and inflammatory PT cells to mediate renal inflammation. Our data also implicate TNF superfamily members in autocrine and paracrine tubular signalling as *TNF* and *TNFSF10* (TRAIL) and their receptors *TNFRSF1A* and *TNFRSF10A,B,D* are all highly expressed in inflammatory PT cells (Fig. 5k, S17c). This may initiate a feed-forward loop to propagate the inflammatory phenotype along the nephron resulting in clustering of inflammatory tubular cells within the same nephron profile (Figs. 2k and 4c–e).

In summary, our integrated ligand-receptor and spatial analysis identifies multiple mechanisms by which inflammatory tubular cells cross-talk with adjacent cells to orchestrate the fibrotic niche.

## Loss of HNF4A network may promote irreversible loss of PT cell identity

We next sought to characterize the epigenetic programming that mediates the transition from healthy tubular cells to this inflammatory

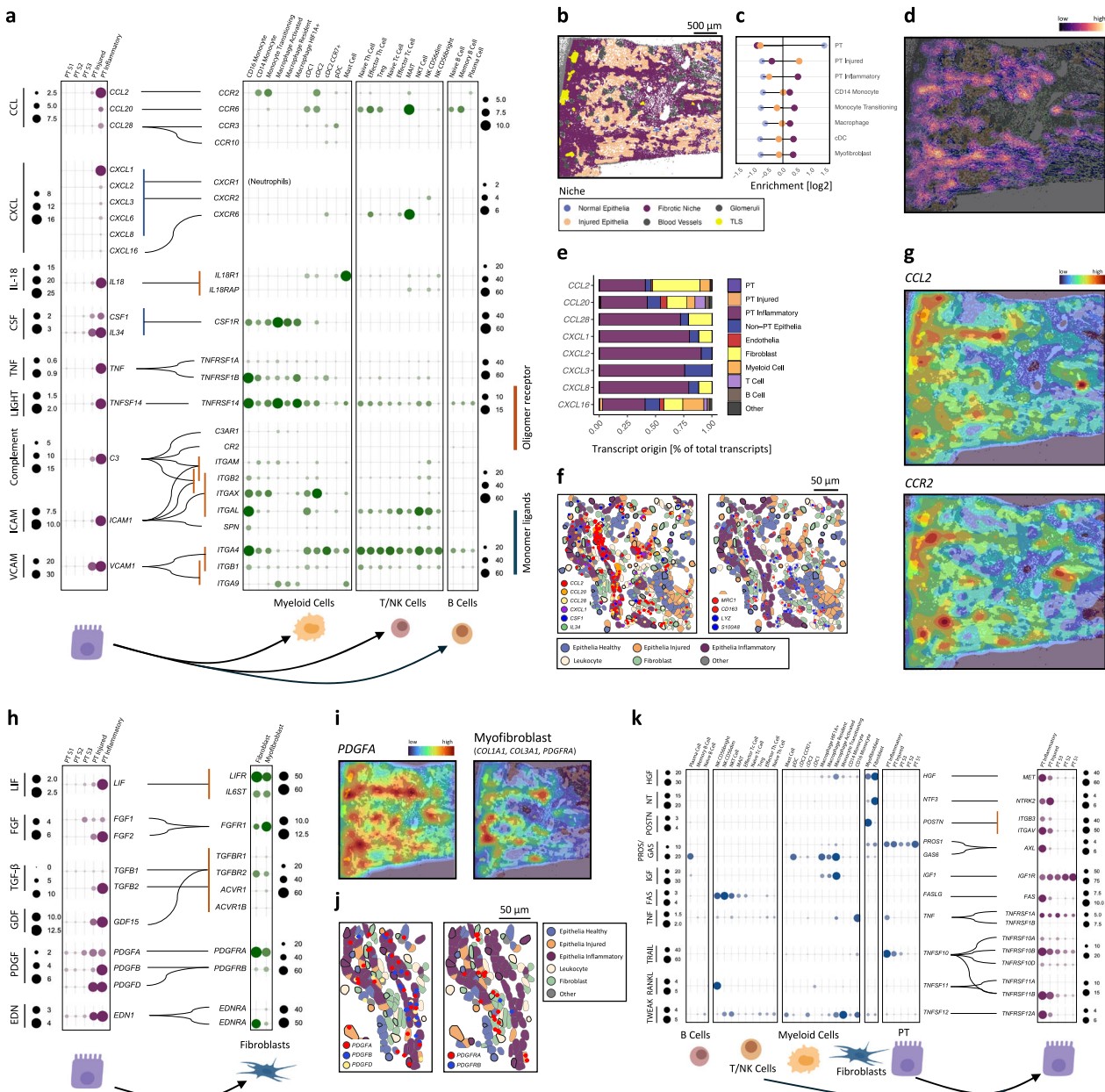

**Fig. 5 | Intercellular signalling mediating immune recruitment and fibroblast activation. a** Dot plots showing expression of ligands in PT clusters (left columns) and respective receptors in immune subclusters (right columns). Dot size is scaled by the fraction of cells expressing the gene and colour by the average (log-scale) scaled expression values. **b** Representative image of an obstructed nephrectomy sample analysed on the CosMx 6000-plex platform. Cells are coloured according to niche assignment. **c** Barplot showing the log2-fold enrichment ratio of observed cell type abundance compared to random tissue distribution within each niche, coloured as per legend in (**b**). **d** Density map indicating the inflammatory PT cell distribution in the same tissue section as (**b**). **e** Barplot showing the contribution of individual cell types to total cytokine transcripts in the fibrotic niche, presented as proportion of total detected transcripts. **f** High-resolution image of spatial localisation of cytokines (left), monocyte-associated (blue) and macrophage-associated

(red) transcripts in the same frame (right). Cells are coloured by cell lineage and epithelial activation state. **g** Density maps showing the spatial distribution of *CCL2* (top) and *CCR2* (bottom) transcripts in the same tissue section as (**b**). **h** Dot plots showing ligands expressed by in PT clusters (left column) and respective receptors in (myo)fibroblasts (right columns). Dot size is scaled by the fraction of cells expressing the gene and colour by the average (log-scale) scaled expression values. **i** Density maps showing the spatial distributions of indicated myofibroblast transcripts in same section as (**b**). **j** High-resolution image of the spatial location of platelet-derived growth factor (PDGF) ligands (left) and receptors (right). Cells are coloured by cell lineage and epithelial activation state. **k** Dotplots showing the expression of ligands (left columns) and upregulated receptors in PT cell subsets (right columns). The dot size is scaled by the fraction of cells expressing the gene and the colour by the average (log-scale) expression.

phenotype. In order to infer transcription factor (TF) dynamics, we leveraged the joint snRNA-seq and ATAC-seq modalities to identify correlations between chromatin accessibility at TF binding sites located within cis-regulatory elements (CREs) and adjacent gene expression and construct gene regulatory networks (Fig. 6a, b, Supplementary Fig. S18.19, Supplementary Data 20–22).

During transition from health to the inflammatory state, PT cells down-regulated genes integral to tubular cell function such as *CUBN* and *SLC34A1* (Fig. 2c); therefore, we asked which TFs govern expression of these down-regulated genes. We linked these genes to their regulatory elements by identifying adjacent CREs with reduced accessibility correlating with lower gene expression. TF binding sites

present within these CREs included those previously implicated in maintaining PT cell health, such as HNF1-α and HNF4-α (Fig. 6c, e, Supplementary Data 20)[10,28]. In order to infer how these TFs co-operate to regulate gene expression, we compared their target genes, observing that TFs broadly segregated into two modules (Fig. 6d), with the target gene expression in module A markedly down-regulated in PT cells during the trajectory from health towards an inflammatory phenotype (Fig. 6f). All other transcription factors in module A, but not module B, had linked CREs containing HNF4-α motifs (Fig. 6g), suggesting that HNF4-α may act as a master-regulator of the network. Hence, loss of HNF4-α activity, as confirmed by progressive reduction of physical occupancy at HNF4-α binding sites during transition from healthy to inflammatory PT cells, (Fig. 6h) may also reduce gene expression of its co-TFs in module A. Our data are consistent with functional studies, as *Hnf4a* knockout in renal tubular cells in mice inhibits maturation of the proximal tubule leading to Fanconi syndrome[29] and *HNF4A* knockout in human kidney organoids reduces PT cell maturation and inhibits expression of genes key to PT cell functions, including ion transport, endocytosis and the brush border[30]. The genes more highly expressed in WT versus *HNF4A* knockout organoids correlated closely with our predicted HNF4A target genes (Supplementary Fig. S20a). Furthermore, in obstructed kidneys these genes were enriched in healthy PT cells, with reduced expression in injured/inflamed PT cells (Supplementary Fig. S20), suggesting loss of HNF4A activity may facilitate transition to an inflamed PT cell phenotype.

To understand the dynamics of HNF-α network disruption following kidney injury, we examined the expression of HNF4-α target genes in PT cells in a time course following murine ischaemia-reperfusion injury (Fig. 6i). HNF4-α target genes were down-regulated within 4 h of ischaemic injury and were restored within 48 h in PT cells that recovered to a healthy phenotype. Conversely, HNF4-α target genes remained persistently low in inflammatory PT cells through to 6 weeks after injury, suggesting failure of the HNF4-α network to recover.

Finally, we addressed why the HNF4-α network remained persistently disrupted after injury. We predicted that HNF4-α bound CREs adjacent to its own gene locus (Fig. 6g) and this was confirmed by previous CUT&RUN analysis showing HNF4-α occupancy at multiple CREs at the *HNF4A* locus in the renal cortex of healthy individuals (Fig. 6j)[30]. During the trajectory from health towards the inflammatory phenotype there was progressive loss of chromatin accessibility at these HNF4-α binding sites coinciding with loss of *HNF4A* gene expression (Fig. 6j). Hence, HNF4-α regulates its own expression in a feed-forward loop, which may be disrupted after depletion of *HNF4A* mRNA and protein in the nucleus during sustained injury. Collectively, these data suggest that disruption of the HNF4-α network contributes to loss of PT identify (Fig. 6k).

## AP-1 promotes transition towards inflammatory tubular cell phenotype

Next, we sought to determine which transcription factors facilitate re-programming of PT cells towards an inflammatory phenotype by activating the injury-related gene expression program (Fig. 2c, g). To this end, we sought TFs for which there was an increased TF score (signature score of target CRE accessibility and target gene expression) combined with increased expression of the TF mRNA in injured and inflammatory PT cells (Fig. 7a). Amongst the most up-regulated TFs were SOX4, KLF6, ELF3 which have previously been implicated in promoting an injured tubular phenotype[7] as well as motifs related to the AP-1 and NF-κβ family of TFs. (Fig. 7a). Next, we asked what the regulatory impact each TF has on gene expression, grouping TF families with similar motifs or which bind as heterodimers (e.g. AP-1 family). Amongst the enriched TFs in the modules gaining accessibility mid-to-late in the trajectory were AP-1 family TFs with motif binding

sites located in ~20% of accessible CREs and NF-κβ1 which was more strongly enriched in late CREs (Fig. 7b, c). Supporting this is a progressively increased TF score in the trajectory from healthy to inflammatory PTs (Fig. 7d) and the highest level of motif occupancy in inflammatory PT as determined by footprinting analysis (Fig. 7e). To confirm these in silico findings experimentally, we stimulated primary renal epithelial cells (RPTECs) with TNF, which is central to CKD progression[31,32] and which activates both the AP-1 and NF-κβ pathways[33,34], and performed a CUT&RUN assay to determine binding sites of NF-κβ1 and JUN (Fig. 7f, Supplementary Fig. S21). We overlapped CUT&RUN peaks with the open chromatin regions in healthy and obstructed kidneys, observing an increase of accessibility during transition from healthy to inflammatory PT cells in a majority of the overlapping regions (Fig. 7g). Furthermore, we found AP-1 and NF-κβ enriched in the respective peak sets confirming increased accessibility at binding sites and physical occupancy of both TFs in inflamed PT cells (Supplementary Fig. S21).

Representative plots of chromatin accessibility at genomic loci of archetypal inflammatory PT cell markers demonstrate highly gene-specific activation patterns (Fig. 7g). For example, accessibility at multiple NF-κβ1 binding sites at the *CCL2* locus is gained by both injured and inflammatory PT cells, while accessibility at the *TNF* locus is relatively restricted to inflammatory PT cells at a single NF-κβ1 binding site. AP-1 has previously been implicated with senescence induction[35] and we observed JUN directly binding to an enhancer element in the *CDKN1A* gene 20 kb upstream of the transcription start site (TSS) which is predominantly accessible in inflammatory PT cells, while the remaining regulatory elements show no changes in accessibility (Fig. 7g). Similarly, four CREs with highest accessibility in inflammatory PTs are observed in proximity to the *HDAC9* locus, two of which bind JUN. Looking at the gene regulatory network more broadly, we predict both distinct and overlapping target genes for NF-κβ1 and JUN (Fig. 7h). For example, in addition to key senescence genes, AP-1 is predicted to activate injury markers such as *HAVCR1* and *DCDC2*. In contrast, NF-κβ1 is essential to activate many pro-inflammatory genes including multiple chemokines, suggesting both AP-1 and NF-κβ1 have intertwined but distinct regulatory roles in promoting the inflammatory phenotype.

## Inhibition of AP-1 activity ameliorates fibrosis following ischaemia-reperfusion injury in mice

As we identified multiple parallel intercellular signalling pathways between inflammatory tubular epithelial cells, leucocytes and myofibroblasts (Fig. 5), we postulated that redundancy may reduce the efficacy of targeting a single ligand-receptor pathway. Therefore, we assessed whether inhibiting core transcriptional regulators of the inflammatory tubular phenotype, such as AP-1, could reduce the severity of renal inflammation and fibrosis. In a murine model of AKI to CKD transition, we targeted AP-1 using T5224, which inhibits binding of c-Fos and c-Jun to DNA thereby preventing transcription factor activity[36]. We administered 10 mg/kg of T5224 or DMSO vehicle by daily gavage for 2 weeks beginning 3 days following unilateral ischaemia-reperfusion injury (IRI) or sham surgery (Fig. 8a). Administration of T5224 ameliorated the reduction in renal mass following IRI (Fig. 8b) and the severity of inflammation and fibrosis (Fig. 8c–e). T5224 also reduced expression of *Arg1*, which we previously identified as a marker of infiltrating monocytes transitioning to a pro-inflammatory phenotype[15]. To assess the effect of T5224 on acquisition of the inflammatory tubular phenotype, we quantified the number of KIM-1⁺ injured tubular cells that also express VCAM1, as unlike in human kidney disease, in murine models VCAM1 expression is more specific to the maladaptive inflammatory PT cell phenotype (Supplementary Fig. S10d). The proportion of KIM1⁺VCAM1⁺ inflammatory PT cells increased following IRI, but was reduced in mice treated with T5224 (Fig. 8f). Furthermore, the expression of genes

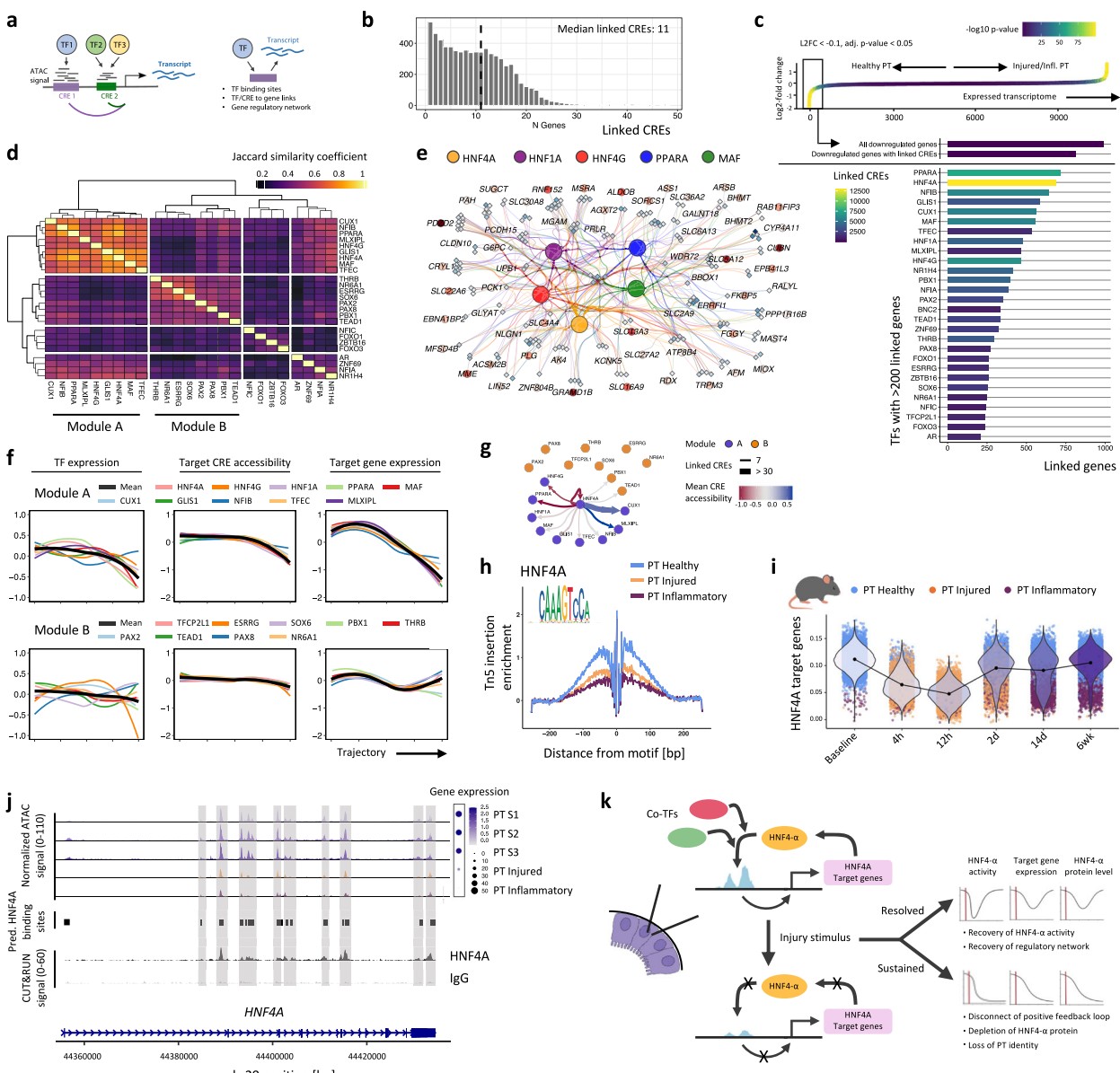

**Fig. 6 | Collapse of an HNF4A-driven gene-regulatory network enables loss of proximal tubule identity. a** Transcription factor (TF) to target gene linkage is inferred when there is high correlation between gene expression and chromatin accessibility in cis-regulatory elements (CREs) harbouring the relevant TF binding motif. **b** Histogram showing the number of CREs linked to each gene. **c** Top panel: genes down-regulated (L2FC < -0.1, adj. *p*-value < 0.05) in injured/inflammatory compared with healthy PT cells (rectangular box). Bottom panel: bar graph showing TFs ordered by the number of down-regulated target genes. Colour indicates the number of CREs linked to down-regulated genes harbouring respective TF motifs. **d** Heatmap showing the Jaccard similarity coefficient between pairs of TFs, with 1 indicating a full overlap and 0 no overlap of target genes. **e** Gene-regulatory network for a subset of module A TFs. CREs are shown as diamonds with darker blue colour indicating decreased accessibility. Target genes are shown as circles with darker red colour indicating stronger down-regulation. **f** Changes in module A (top) and module B (bottom) TF activity along the pseudotime trajectory from healthy to

inflammatory PT cells (Fig. 2f). Left: expression of genes encoding TFs. Middle: accessibility score for TF-target CREs. Right: expression of TF target genes. **g** HNF4-α binding sites in CREs linked to TFs in module A and B. Edge width indicates the number of bound CREs and colour the mean accessibility change (negative values indicate reduced accessibility in injured/inflammatory compared with healthy PT cells). **h** HNF4-α motif footprint in PT cell states. **i** Violin plots of HNF4-α target gene expression in PT cells (dots coloured according to cell phenotypes) following ischaemia-reperfusion injury in mice (Fig. 2i). **j** Upper: chromatin accessibility in healthy, injured and inflammatory PT cells at the *HNF4A* locus with corresponding *HNF4A* mRNA expression (right, dot size scaled by the proportion of cells expressing the gene and coloured by the average (log-scale) expression). Middle: regions harbouring HNF4-α motifs. Lower: HNF4-α CUT&RUN signal in adult human kidney[30]. **k** Sustained injury disrupts the HNF4A auto-regulatory loop, depleting *HNF4A* mRNA and protein leading to loss of PT identity.

characteristic of inflammatory tubular cells including *Icam1*, *Vcam1* and *Pdgfa* was also reduced in animals treated with T5224 highlighting potential mechanisms for the reduced leucocyte and myofibroblast accumulation (Fig. 8g). This is in keeping with CUT&RUN analysis in TNF-treated RPTECs, which demonstrated that JUN bound to CREs adjacent to the *HAVCR1*, *VCAM1*, *ICAM1* and *PDGFA* genes (Fig. 8h).

Importantly, these CREs were located in chromatin regions that were progressively more accessible in PT cells during transition from health to the inflammatory phenotype in human kidney disease. Taken together, the data confirm that inhibition of AP-1 activity using T5224 inhibited acquisition of the inflammatory tubular cell phenotype and transition from acute ischaemic kidney injury to renal fibrosis.

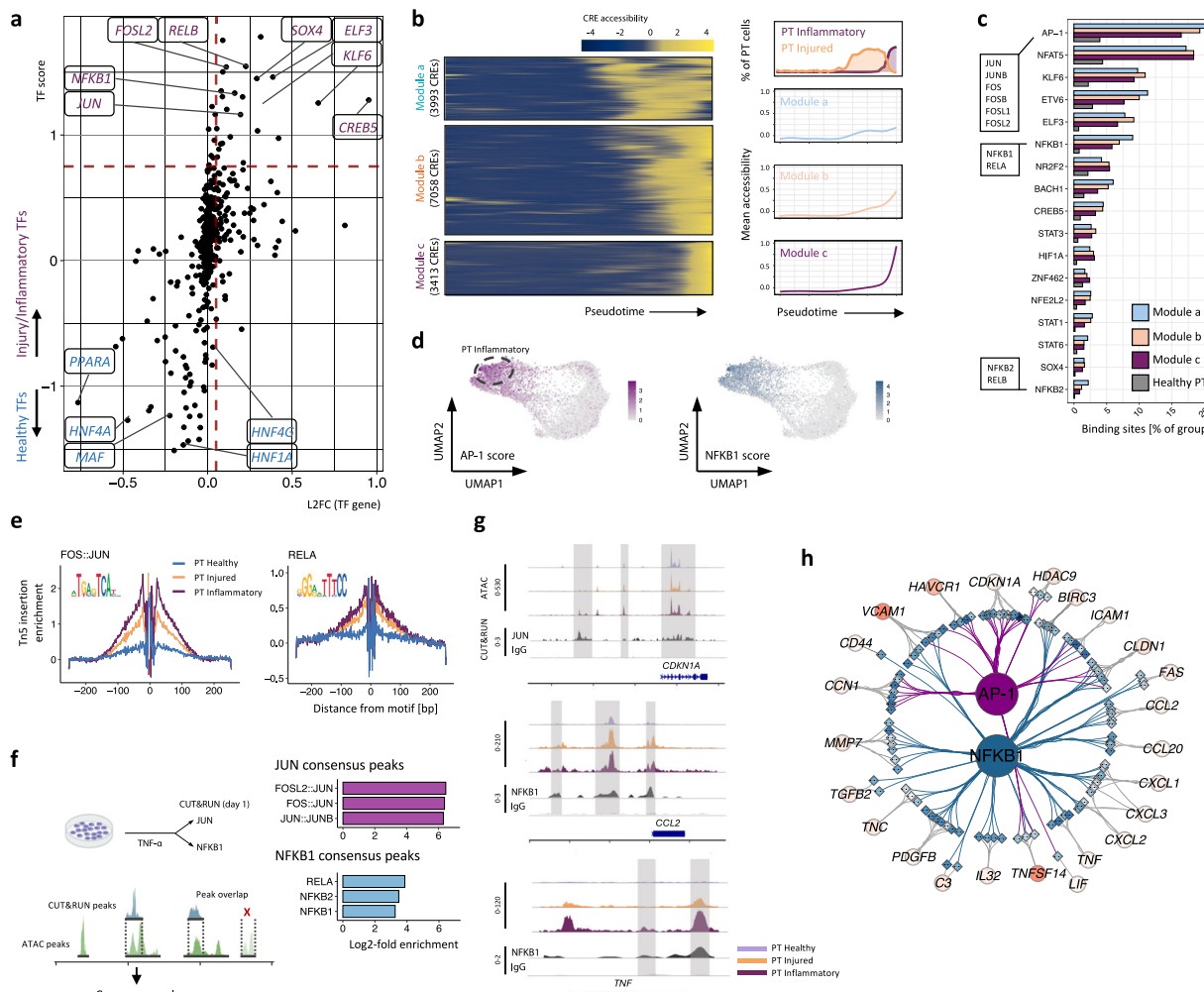

**Fig. 7 | Transcription factors associated with inflammatory programming of proximal tubule cells. a** TF gene expression plotted against the TF score (mean of target gene and target CRE signature scores) highlights TFs enriched in healthy and injured/inflammatory PT cells. **b** Chromatin accessibility dynamics along the trajectory from healthy to inflammatory PT cells (Fig. 2f). Left panel: CREs linked to upregulated genes (L2FC > 0.1, adj. p-value < 0.05) clustered into modules with increased accessibility early, intermediate and late in the trajectory. Right panel: Smoothed graphs summarizing the mean accessibility for CREs in each module, with the percentage of injured and inflammatory PT cells above. **c** Proportions of CREs with putative binding sites for the respective TF family, grouped by module. **d** UMAP (Fig. 2a) of the AP-1 and NF-κβ1 TF score in PT cells. **e** TF footprint of JUN:FOS and RELA motifs in healthy, injured and inflammatory PT cells. **f** Left panel:

intersection of chromosomal regions binding JUN or NF-κβ1 by CUT&RUN assay in renal proximal tubular cells (RPTECs) following TNF administration with differentially accessible chromatin in inflammatory PT cells. Right panel: Motif enrichment ratio in regions of intersection. **g** Chromatin profile and CUT&RUN signal tracks at archetypal inflammatory PT genes with AP-1 or NF-κβ1 target CREs. Upper tracks: The ATAC signal in healthy and injured/inflammatory PT cells. Lower track: The JUN or NF-κβ1 CUT&RUN peaks indicating the physical presence of the TF at genomic loci in TNF-treated RPTECs. **h** Visualisation of the predicted regulatory links between AP-1 or NF-κβ1 and selected genes upregulated in inflammatory PT cells (Fig. 2c). CREs are shown as diamonds (darker blue colour indicating increased accessibility) and target genes as circles (darker red colour indicating greater up-regulation). TF to CRE edges are coloured by the TF bound to the CRE.

## Senolytic therapies promote depletion of the inflammatory tubular cell phenotype

We had previously identified that inflammatory PT cells persisted despite cessation of injury and up-regulated *CKDN1A* and senescence-associated genes, suggesting that they had adopted a senescent phenotype (Fig. 2c, Supplementary Fig. S11). JUN bound to AP-1 sites in the *CKDN1A* promoter in inflamed RPTECs (Fig. 7g) and AP-1 inhibition reduced *Ckdn1a* expression (Fig. 8e), suggesting that AP-1 may promote transition to this senescent phenotype. However, we also wanted to assess whether depletion of cells that had already become senescent was associated with a reduction in the inflammatory tubular cell phenotype. We determined that expression of the anti-apoptotic gene *BCL2* was increased in inflammatory tubular cells, which may render them resistant to apoptosis and enable them to persist despite cessation of injury (Fig. 8i). Hence, we assessed whether depletion of

senescent cells using the BCL2/w/xL inhibitor ABT-263, a senolytic compound known to promote renal repair[37,38], also reduced expression of genes consistent with an inflammatory tubular cell phenotype (Fig. 8j). In the reversible ureteric obstruction model, archetypal inflammatory tubular cell genes (Fig. 2c) were up-regulated in the renal cortex one week following reversal of obstruction (Fig. 8k). In mice that received vehicle, these genes remained up-regulated 5 weeks following reversal of obstruction, indicating persistence of the inflammatory tubular phenotype, whereas administration of ABT-263 markedly reduced their expression (Fig. 8k) and inhibited development of renal fibrosis[38]. Furthermore, ABT-263 reduced the proportion of inflammatory PT cells as inferred from deconvolution of bulk RNA-seq data (Fig. 8l). Hence, senolytic therapies may represent an alternative approach to deplete inflammatory tubular cells that have developed following kidney injury.

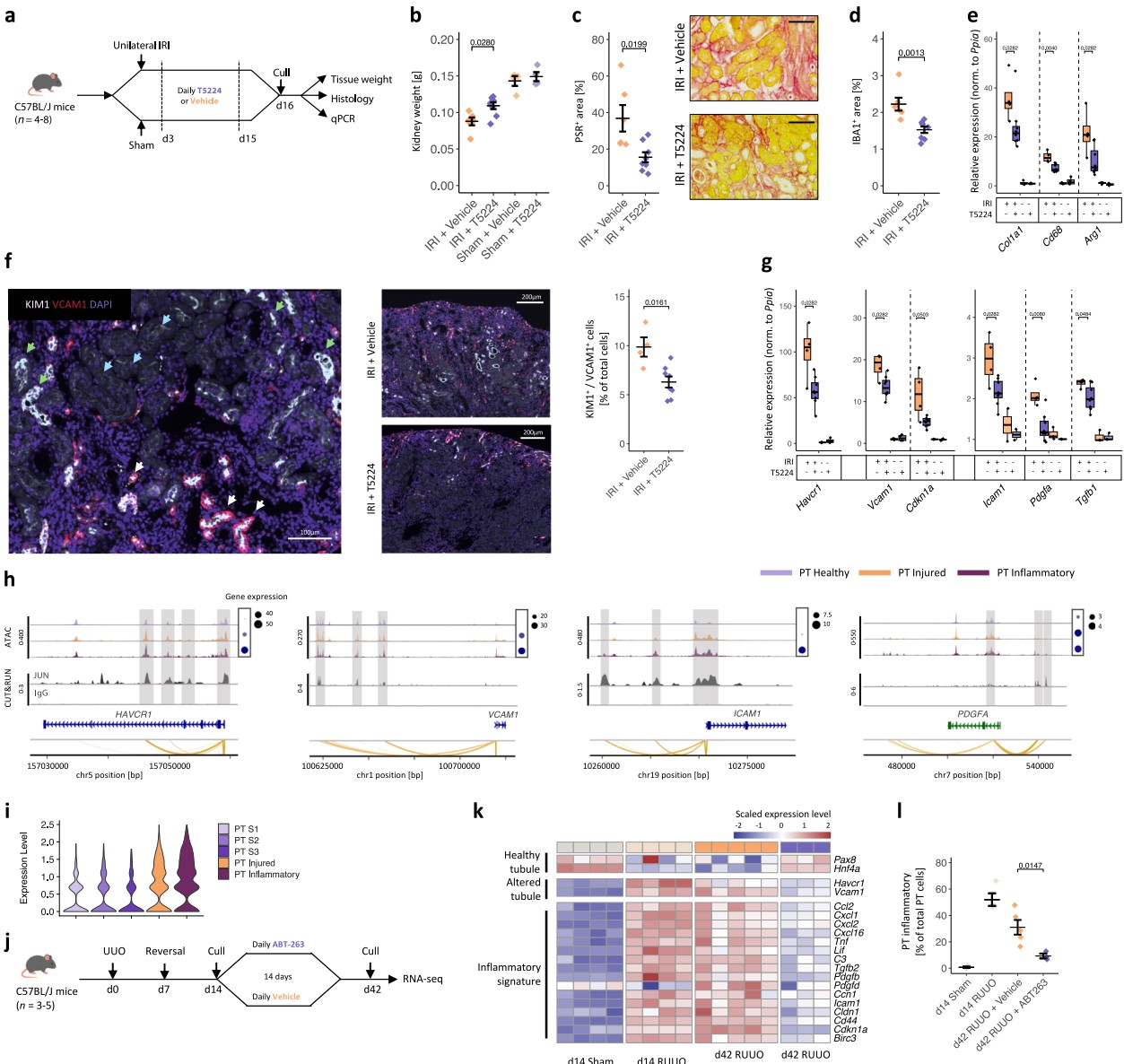

**Fig. 8 | Targeting inflammatory PT cells ameliorates fibrosis in mouse models of AKI to CKD transition. a** Experimental schema: mice underwent IRI and were administered T5524 (*n* = 8) or vehicle (*n* = 6) or underwent sham surgery and administered T5224 or vehicle (*n* = 4 for both groups). Kidney weights (**b**) and percentage of renal cortex staining for picrosirius red (PSR) with representative images (**c**) or the macrophage marker IBA1 (**d**). **e** Expression of fibrosis and myeloid cell genes in renal cortex. **f** Left: immunofluorescence for KIM1/VCAM1. White arrows: KIM1⁺/VCAM1⁺ inflammatory tubules; green arrows: KIM1⁺/VCAM1⁻ injured tubules; blue arrows: healthy KIM1⁻/VCAM1⁻ tubules. Middle: representative images from T5224 and vehicle-treated mice. Right: KIM1⁺/VCAM1⁻ or KIM1⁺/VCAM1⁺ cells in proportion to total PT cells. **g** Expression of injured and inflammatory PT cell genes in renal cortex. **h** Top tracks: The chromatin accessibility at AP-1 target genes in human PT cell subsets with corresponding gene expression indicated by dot plots

(right). Bottom tracks: JUN binding assessed by CUT&RUN in TNF-treated RPTECs (Fig. 6f). ATAC peaks that correlate with gene expression are linked to the TSS (yellow line). **i** Violin plot of *BCL2* expression in human PT subsets (log scale). **j** Experimental schema: Mice underwent unilateral ureteral obstruction (UUO), which was reversed after 7 days followed by 14 daily gavages of BCL2 inhibitor ABT-263 or vehicle[38]. **k** Heatmap of scaled gene expression in renal cortex from mice undergoing sham surgery, at 7 days post reversal of ureteric obstruction (R-UUO, d14) or after R-UUO and then treatment with ABT-263 or vehicle. **l** Inflammatory PT cells as a proportion of total PT cells derived from deconvolution of bulk RNA-seq of renal cortex from animals undergoing sham surgery (*n* = 4), 7 days after reversal of UUO (d14 R-UUO, *n* = 4), or 35 days after reversal of UUO with administration of ABT (*n* = 3) or vehicle (*n* = 5). For all data, values are means ± SEM, with analyses performed using Wilcoxon rank-sum test.

## Discussion

Our multiomic dataset highlights that in response to injury, PT cells adopt a spectrum of cell states. These range from a reversible, early injury phenotype characterised by expression of *HAVCR1* and *VCAM1* and broadly analogous to 'adaptive' tubular cells identified in the KPMP cell atlas, to a small subset of persisting cells, which additionally express *ICAM1* and pro-inflammatory, pro-fibrotic genes consistent with a role in driving interstitial fibrosis. Similar inflammatory tubular

epithelial phenotypes were observed in other regions of the nephron, in particular the Loop of Henle.

We apply high-plex spatial molecular imaging using the CosMx platform to human kidney disease, leveraging the single cell resolution to determine that it is specifically the *ICAM1*⁺ inflammatory PT cells that localize to the fibrotic niche. Their tight association with the fibrotic niche as well as their deleterious transcriptome suggests an active role in immune cell recruitment and fibroblast activation and indeed, their

abundance correlates with the severity of fibrosis. Conversely, injured *VCAM1*+ PT cells that did not co-express *ICAM1* lacked expression of chemokines and pro-fibrotic factors and were not strongly associated with the fibrotic niche, suggesting that they may represent an intermediate, potentially reversible injury phenotype. Our ligand-receptor analysis identified multiple signalling pathways by which the inflammatory PT cells mediate leucocyte chemotaxis and fibroblast-to-myofibroblast activation and our spatial analysis confirmed that the ligands and receptors are in close proximity to enable effective signalling. Inhibition of several of these ligand-receptor pairs ameliorates murine kidney disease[13,24,27,39,40] and our data suggest that they offer additional therapeutic opportunities in human disease. While our study and that of the KPMP focused on diseases not typically considered to be immune-mediated, such as obstructive, diabetic and hypertensive nephropathy, the presence of ICAM1+ tubular cells in glomerulonephritis[41] and lupus nephritis[42], suggests that it may represent a common mechanism driving tubulointerstitial fibrosis regardless of aetiology. Further studies using standardized methodologies across a wide spectrum of kidney diseases and stages of disease progression will be required to identify disease- and stage-specific pathogenic cell phenotypes to enable targeted therapies. In particular, biopsies from early disease may be helpful in determining the sequence of events that initiate development of the fibrotic niche; for example, whether acquisition of the inflammatory phenotype following tubular injury initiates leucocyte recruitment or is secondary to cytokines derived from infiltrating leucocytes.

The inflammatory PT cells are broadly analogous to *Vcam1*+ 'failed repair' cells previously observed following ischaemic renal injury in mice[12,13]. While *Vcam1* is specific to failed repair PT cells in mice, we have found that in human disease *VCAM1* expression is common to both injured and inflammatory PT cells, while *ICAM1* specifically labels the inflammatory PT cells (Supplementary Fig. S22). We developed an antibody panel that could readily be utilized to quantify the burden of VCAM1+ICAM1+ inflammatory PT cells in kidney biopsy tissue, facilitating identification of patients who may benefit from targeted therapies such as AP-1 inhibition or senolytics. Furthermore, ICAM1 may contribute to disease pathogenesis by propagating the inflammatory response as it localizes to the luminal membrane of inflammatory PT cells and enables tethering myeloid cells present in the urine. Indeed, expression of ICAM1 in the proximal tubules has been associated with progression to chronic and end-stage kidney disease in patients with lupus nephritis[42] and serum ICAM-1 levels predict CKD and decline of kidney function in the general population[43].

We leveraged the simultaneous single-cell and ATAC-sequencing datasets to characterise the key transcription factors that promote tubular cell integrity and identified AP-1 as a key mediator of transition to the inflammatory state. While AP-1 inhibition has previously been shown to prevent endotoxin-injured kidney injury in pre-clinical models[44], our data suggests that it may have a wider role in preventing transition from AKI to chronic inflammation and fibrosis. AP-1 has also been implicated as a key regulator of cellular senescence, with inflammatory PT cells being enriched with senescence-associated transcripts. Depletion of senescent cells after kidney injury reduced expression of the inflammatory tubular cell signature, suggesting that senolytic therapies may be an alternative treatment for patients where the inflammatory tubular phenotype is well established and irreversible, which is important as many patients present with advanced CKD.

In summary, we have generated a comprehensive multiomic atlas of the cellular heterogeneity in human kidney disease and identified druggable signalling pathways by which an ICAM1+ inflammatory tubular cell subset may mediate tubulointerstitial inflammation and fibrosis. Furthermore, we confirmed that inhibition of AP-1 activity and senolytic therapies reduce progression of inflammation and fibrosis following experimental AKI and warrant further research as a

treatment to prevent AKI to CKD transition and progression of CKD in patients with kidney disease.

## Methods

### Ethics
Our studies comply with all relevant ethical regulations for human and animal research. Patients provided written informed consent for use of their surplus tissue for our research purposes. The study was approved by the steering committee of the National Research Scotland Lothian Bioresource, REC 20/ES/0061, following separate applications for use of nephrectomy (SR1651) and biopsy (SR1887) specimens.

All animal procedures had prior approval of the Animal Ethics Committee, University of Edinburgh and were conducted in accordance with the United Kingdom Animals Scientific Procedures Act 1986 and the ARRIVE guidelines.

### Patient recruitment
For samples used in the multiome analysis and CosMx 6k panel analysis, patients undergoing radical nephro-ureterectomy or radical nephrectomy at the Western General Hospital, Edinburgh were recruited by the consultant urologist performing the procedure. We recruited patients with renal cell carcinoma ($n = 6$) or oncocytoma ($n = 1$) and normal renal function (eGFR >75 ml/min/1.73 m$^2$) as controls and in addition we recruited patients with transitional cell carcinoma of the ureter ($n = 5$) leading to ureteric obstruction as determined by the presence of hydronephrosis on CT imaging.

For samples used in the CosMx 1K-panel analysis, we utilised surplus formalin-fixed, paraffin-embedded tissue from kidney biopsies from patients with IgA nephropathy ($n = 6$) or minimal change disease ($n = 3$) that were performed for clinical purposes. In addition, we employed wedge biopsies from nephrectomy specimens removed due to advanced pyelonephritis ($n = 4$).

Renal function at the time of nephrectomy or biopsy and during follow-up was estimated from serum creatinine results provided by the clinical lab by applying the CKD-EPI equation without correction for race[45]. Urinary protein:creatinine results were provided by the clinical lab.

### Animal experiment
24 male C57BL/6JCrl mice, aged 6–8 weeks were purchased from Charles River Laboratories and housed in a pathogen-free environment at the University of Edinburgh animal facility at 50% humidity and 22–26 °C with 12 h light: dark cycles starting at 07:00 local time.

IRI surgery was performed as previously described[46]. Briefly, a posterior flank incision was made and the left renal pedicle identified and clipped using an atraumatic clamp for 15 min. During the ischemic period, body temperature was maintained at 37 °C using a heating blanket with homeostatic control (Harvard Apparatus) via a rectal temperature probe. The clamp was then removed, the peritoneum closed with 5/0 suture, and the skin closed with clips. In sham surgery, the mice underwent flank incision, however the renal pedicle was not clamped.

T5224 was reconstituted in dimethyl sulfoxide (DMSO) and diluted 1:20 in sunflower oil. 3 days post-IRI mice in the respective group were given 10 mg/kg T5224 (APExBIO, B4664) by gavage daily for 12 days. Animals were maintained for 15 days post-IRI before tissue harvest. IRI or respective control kidneys were weighed, and one third was fixed for 24 h in 10% neutral buffered formalin with the remainder snap-frozen in dry ice.

All mice were operated upon before assignment to T5224 or vehicle on a whole cage basis (rotating cages for each mouse, with the operator unaware of subsequent therapy) with 8 mice in each treatment group. The experiment was powered to detect a 20% reduction in fibrosis between the groups. In all subsequent analyses, the investigators were blinded to the treatment group to which each animal was assigned.

## Histology

After fixation and paraffin embedding, human and mouse tissues were sectioned at 4 μm depth from FFPE embedded tissue blocks and adhered to Superfrost Plus slides (VWR, 631-0108). Sections were deparaffinized two times 5 min in xylene and rehydrated in 100%, 90% and 70% ethanol for 2 min each and placed in distilled water. PSR staining was performed using the Picro Sirius Red Stain Kit (Abcam, ab150681) following manufacturer guidelines. After rehydration of tissue sections, the tissue was covered picro sirius red solution for 60 min. Slides were rinsed in acetic acid solution and dehydrated in absolute ethanol before mounting. For macrophage quantification, murine kidney sections were incubated with the avidin/biotin blocking kit (Vector Laboratories, SP2001) and blocked with serum-free protein block (Dako, X0909) before incubation overnight at 4 °C with IBA1 antibody (Invitrogen, PA5-27436) diluted 1:250 in antibody diluent (Dako UK Ltd, S202230). After washing, the sections were incubated with polyclonal goat anti-rabbit biotinylated secondary antibody (DAKO, E0432, 1:300 dilution) for 30 min at room temperature. Vectastain RTU ABC Reagent (Vector Laboratories, PK7100) was then applied, followed by incubation with the DAB+ Substrate Chromogen System (Dako, K3468), and then counterstaining with hematoxylin before dehydration and mounting with Pertex mounting medium (Histolab Products AB, 3808707E).

## Immunofluorescence

For immunofluoresence, 4 mm FFPE sections were rehydrated as above and then underwent antigen retrieval by microwaving (20 min, 60% microwave power) in Tris-EDTA buffer containing 1.21 g Tris Base (Sigma, T6066) and 0.37 g EDTA (VWR, 20302) in 1000 ml distilled water. Sections were blocked using BLOXALL Endogenous Blocking Solution (Vector Laboratories, SP-6000-100) for 10 min at room temperature and Protein Block (Agilent, X090930-2). Antigens were then detected using Opal fluorophores (NEL861001KT, Akoya Biosciences). For this the primary antibody was incubated for 1 h at room temperature with the antibodies and opal dye combinations as provided in Supplementary Data 23. Next, a HRP-labelled secondary antibody (anti-rabbit HRP or dual anti-mouse and anti-rabbit HRP, Akoya Biosciences or ImmPRESS HRP Horse Anti-Goat IgG Polymer Detection Kit (MP-7405, Vector Laboratories) was incubated for 10 min at room temperature. Lastly, stains were developed by adding the designated Opal fluorophores, diluted 1:100 in Manual Amplification Diluent (FP1498, Akoya Biosciences) for 10 min at room temperature. Subsequently, antigen retrieval was repeated to remove bound antibodies. This procedure was repeated for each antibody until each fluorophore was developed. Last, slides were counterstained with 10X Spectral DAPI (FP1490, Akoya Biosciences) diluted in PBS for 5 min at room temperature before mounting of coverslips.

## Image analysis

PSR stained sections were scanned with a Zeiss AxioScan Z1 slide scanner (Carl Zeiss). Images of human kidneys were reviewed by a consultant pathologist who confirmed the absence of tumour infiltration and provided a semi-quantitative assessment of glomerulosclerosis and tubulointerstitial inflammation and fibrosis. For mouse tissue, PSR staining of renal cortex was quantified in QuPath (v0.4.2). A threshold classifier was used to calculate cortex area with intensities above the thresholds denoting collagen fibres and this was assessed as a percentage of total tissue area. The percentage of total renal cortex that stained with IBA1 was quantified using a minimum threshold for positive staining on ImageJ analysis using a minimum of 5 ×20-magnification images of renal cortex.

For immunofluorescence, slides were imaged using the Vectra Polaris Imaging System (Akoya Biosciences) at ×20 magnification following manufacturer instructions. Spectral unmixing of fluorescence images was performed using the inForm software (v2.7.0, Akoya Biosciences) with default settings.

For analysis of mouse HAVCR1/VCAM1 dual staining the images were loaded into QuPath (v0.5.0). Initially, cortical regions were segmented using polygon annotations. Next, cell segmentation as performed using QuPath's 'Cell detection' function with default settings. To quantify the abundance of HAVCR1$^+$/VCAM1$^+$ cells a Random trees classifier was used to categorise cells into HAVCR1$^+$/VCAM1$^-$, HAVCR1$^-$/VCAM$^+$, HAVCR1$^+$/VCAM1$^+$ and HAVCR1$^-$/VCAM1$^-$ classes based on fluorescence intensities. To achieve this, QuPath's 'Train object classifier' function was used on a set of training images derived from the dataset. The classifier was trained to recognise the respective classes, verified by manual inspection of results. Results of cell counts in cortical regions of each sample were exported and compared using the Student's $t$-test.

To analyse 5-plex immunostaining of human kidney tissue sections, the scanned images were loaded in QuPath (v0.5.0). Initially, images were segmented based on the intensity of PanCK, FAP or CD68 stains. Tubules were segmented using a threshold classifier based on PanCK intensity (non-default settings: resolution = full; threshold = 35, smooth = 0.5, minimum object size = 80 μm$^2$, minimum hole size > 1000 μm$^2$). Where automatic segmentation polygons failed to recapitulate the true tubular structure, polygons were manually adjusted with the goal of obtaining closed polygons for each tubule. FAP$^+$ (non-default settings: resolution = high; threshold = 2.5, smooth = 1, minimum object size = 10 μm$^2$) and CD68$^+$ (non-default settings: resolution = full; threshold = 2, smooth = 0, minimum object size = 4 μm$^2$) segmentation objects were obtained in the same way but were not manually adjusted. Subsequently, features were generated for each annotation, measuring mean fluorescence intensity of in all channels.

Polygons and associated measurement were exported to R and fluorescence measurements were log2 + 1 transformed. Tubule segmentations were pruned to remove tubules with abnormally high CD68 or FAP intensities (mean CD68 > 0.6 or mean FAP > 1.8). VCAM1$^+$ tubules were identified as tubules with mean VCAM1 intensity > 2 (and mean PanCK > 4.8, < 7). VCAM1$^+$/ICAM1$^+$ tubules were defined as VCAM1$^+$ tubules with a mean ICAM1 intensity > 0.6. To compute a UMAP embedding for all segmentation objects principal components were computed based on mean intensity measurements using the prcomp function (with centre = T) and the UMAP was computed using the umap R package (v0.2.10.0) with default settings. Lastly, to compute the area coverage by FAP$^+$ and CD68$^+$ objects in proximity to tubules a buffer area expanding 25 μm from tubule segmentation boundaries was created. This buffer area was intersected with FAP$^+$ and CD68$^+$ polygons and the percentage of buffer area covered by FAP$^+$ and CD68$^+$ objects was computed. The percentage values were grouped using the previous described tubule classes and compared using a Wilcoxon rank-sum test with datapoints representing individual tubule segmentations.

## qPCR

Total RNA was extracted from kidney tissue using the miRNeasy Mini kit (Qiagen, 217004), following the kit instructions. cDNA for quantitative PCR was synthesised from the extracted RNA using the High-Capacity cDNA Reverse Transcription kit (Applied Biosystems, 4368814). Quantitative PCR was performed using TaqMan Universal Master Mix II (Applied Biosystems, 4440040) and TaqMan Gene Expression Assay-specific primers (Supplementary Data 24) and normalised to Peptidylprolyl isomerase A (*Ppia*) expression.

## DNA extraction and genotyping

DNA was extracted from approximately 25 mg of nephrectomy tissue using the DNeasy Blood & Tissue Kit (Qiagen, 69504) according to the manufacturer instructions. Briefly, tissue was cut into small pieces and

incubated with proteinase K for 3 hours at 56 °C. Genomic DNA was purified using a DNeasy Mini spin column, washed and eluted with TE buffer. 200 ng of genomic DNA per sample was used to genotype samples using an Infinium Global Screening Array (Illumina, 20030770) according to manufacturer instructions. Raw genotyping data was converted to variant call format (VCF) using GenomeStudio (Illumina, v2.0.3).

## Single-nucleus multiome RNA and ATAC sequencing

**Nuclei isolation, library preparation and sequencing.** For single-nuclei multiome studies wedge biopsies were taken immediately after nephrectomy from the pole opposite the tumour in those with intra-renal tumours and samples snap-frozen in dry ice and a separate piece was fixed for 24 h in 10% neutral buffered formalin before paraffin embedding for histological analysis. Nuclei were isolated from snap-frozen nephrectomy samples using an adapted 10x Genomics Demonstrated protocol (CG000375, Nuclei Isolation from Complex Tissues for Single Cell Multiome ATAC + Gene Expression Sequencing). Using a scalpel, the tissue was cut into approximately 25 mg of rice-grain sized pieces while avoiding thawing and all further steps were performed at 4 °C. Initially, 150 µl lysis buffer containing 10 mM Tris-HCl (Sigma-Aldrich, T2194), 10 mM NaCl (Sigma-Aldrich, 59222 C), 3 mM $MgCl_2$ (Sigma-Aldrich, M1028), 0.05% NP40 Substitute (Sigma-Aldrich, 74385), 1 mM DTT (Sigma-Aldrich, 646563) and 1 U/µl Protector RNase inhibitor (Sigma-Aldrich, 3335402001) in Nuclease-free Water (Invitrogen, 10429224) was added and tissue was homogenised using a sterile pestle (Fisher Scientific, 12-141-368). Once homogenised another 850 µl lysis buffer was added, and tubes were placed on ice for 5 min. The suspension was strained using a 40 µl filter (Cambridge Bioscience, 43-10040-60), centrifuged (500 × g, 5 min, 4 °C) and 1 ml 2% BSA (Miltenyi Biotec,130-091-376) in phosphate-buffered saline (PBS) (ThermoFisher, 10010023) was added without disrupting the pellet. After 5 minutes nuclei were resuspended, centrifuged (500 × g, 5 min, 4 °C) and finally resuspended in 300 µl of 2% BSA in PBS.

Nuclei were stained with 10 µl of 7-AAD viability staining solution (Thermo Fisher Scientific, 00-6993-50) and sorted on a BD FACSAria fusion (Becton Dickinson) cell sorter fitted with a 100 ml nozzle into a tube prepared with 200 µl of 10% BSA in PBS supplemented with 50 µl RNase inhibitor for a post-sorting volume of 2 ml. Between 3 and 5 samples from individual donors were pooled at pre-defined proportions based on counts of 7-AAD+ nuclei (Supplementary Fig. S2b) and a total of 500,000 nuclei were sorted per pool (Supplementary Fig. S2a).

Sorted nuclei were centrifuged (500 × g, 5 min, 4 °C) and resuspended in 100 µl lysis buffer with 0.01% NP40 Substitute, 0.01% Tween-20 (Bio-Rad, 1662404), 0.001% Digitonin (Thermo Fisher Scientific, BN2006) and 2% BSA. After 2 min wash buffer (10 mM Tris-HCl, 10 mM NaCl, 3 mM MgCl2, 2% BSA, 1 mM DTT, 1 U/µl RNase inhibitor, 0.1% Tween-20) was added and nuclei were centrifuged (500 × g, 5 min, 4 °C) and resuspended in 20 µl diluted nuclei buffer (20X Nuclei Buffer, 1 mM DTT, 1 U/µl RNase inhibitor in nuclease-free water).

Nuclei suspensions were counted using a LUNA-FX7 cell counter (Logos Biosystems) and processed immediately using the Chromium Next GEM Single Cell Multiome ATAC + Gene Expression kit (v1.0, 10x Genomics) according to manufacturer protocol (CG000338, User Guide Chromium Next GEM Single Cell Multiome ATAC + Gene Expression). Briefly, per library between 20,000 and 40,000 nuclei were input into the transposition reaction before partitioning into gel beads. After reverse transcription and pre-amplification, aliquots of the pooled transposed DNA and cDNA were used to generate separate ATAC and GEX libraries.

After quantification GEX and ATAC libraries were pooled at molarity ratio of 40:60 respectively and sequenced using an Illumina NovaSeq6000 flow cell (with the read lengths R1:151 bp I1:10 bp I2:24 bp R2:151 bp) for a target yield of approximately 50,000 reads/ cell and 75,000 reads/cell for gene expression and ATAC libraries respectively. In total 6 libraries comprising a total of 7 individual control and 5 UUO individual samples were prepared and analysed.

## Sequencing data processing and donor demultiplexing

BCL files from Illumina sequencing runs were demultiplexed for GEX and ATAC samples using cellranger-arc mkfastq (v2.0.2, 10x Genomics). The base mask Y28n*,I10n*,I10n*,Y151 and --filter-dual-index or Y100n*,I8n*,Y24,Y100n* and --filter-single-index was used to generate GEX or ATAC FASTQ files respectively. Reads were aligned to the precomputed GRCh38 (hg38) reference genome provided by 10x Genomics, and quantification as well as joint cell calling was performed using cellranger-arc count (v2.0.2).

Due to donor pooling, each library included cell barcodes from multiple different samples. To allocate barcodes to their original donor the cellsnp-lite (v1.2.2)[47] and vireo (v0.2.3)[48] packages were used. Barcodes passing cellranger-arc filters were genotyped independently for GEX and ATAC libraries by searching for SNPs present in sequencing reads using cellsnp-lite with the options --minMAF 0.1 (and --UMItag None for ATAC files) using a reference SNP list of 7.4 million SNPs from the 1000 Genomes Project, filtering for a minor allele frequency > 5% (provided by cellsnp-lite). To infer the original donor of each nuclei barcode using genetic variants that segregate between samples, a Bayesian model implemented by vireo was used (with option –N equal to the number of donors in the library). As donors were predicted independently for GEX and ATAC libraries, conflicts in donor assignments arising in a minority of nuclei barcodes were resolved using the following logic: i) barcodes assigned as doublet in either modality or unassigned in both modalities were removed (18.36% of total); ii) barcodes which were unassigned in one modality but not the other were assigned to the donor predicted by the successful modality (1.55% of total); iii) barcodes with conflicting donor assignments were removed (98.68% matching, 1.32% conflicting; % of remaining after previous filters) (Supplementary Fig. S2h).

To link predicted donors with physical samples two parallel approaches were used. Firstly, using the original tissue, genomic DNA was extracted and genotyped using an Illumina an Infinium Global Screening Array as described above. VCF files containing the predicted genotype of each donor-library combination were loaded in R (v4.3.1) and subset to SNPs commonly detected in all libraries. To intersect the predicted genotypes with the SNP array, both sets of SNPs were further reduced to a common set and a mean distance matrix was calculated based on the Hamming distance at each genomic location (with 0 being a homozygous reference allele, 2 a homozygous alternative allele and 1 being a heterozygous allele) (Supplementary Fig. S2k). Secondly, a subset of samples was pooled in multiple libraries creating a distinctive pooling pattern which can be used to identify each sample (i.e. the same sample pooled in two different libraries is expected to have the same genotype, as predicted by vireo) (Supplementary Fig. S2i). VCF files were processed as previously and the mean Hamming distance between each donor-library pair was calculated, showing a lower distance between identical samples used in multiple libraries (Supplementary Fig. S2j).

## Joint dimensionality reduction, clustering and cell annotation

While SNP-based donor demultiplexing allows identification of doublets with distinct genotypes, doublets of nuclei from the same donor remain undetected. Therefore, we used an iterative approach to identify undetected doublets, initially retaining SNP doublets to aid identification. Seurat (v4.4.0)[49] and Signac (v1.11.0)[50] were used to load GEX data using Read10X_h5() and a Seurat object was created using the CreateSeuratObject function without filters. The data was normalised using SCTransform with options vst.flavor = "v2" and principal components were computed using RunPCA with npcs = 100. Different dimensionality reduction and clustering setting were used to compute

different representations of the data with RunUMAP, FindNeighbors (with dims between 1 and 80) and FindClusters (resolution between 0.5 and 4). At each iteration presumed doublet subclusters were marked based on the criteria: i) enrichment of doublets identified based on SNP demultiplexing (presence of SNPs from different genetic backgrounds); ii) increased number of unique molecular identifiers (UMIs) or unique genes per barcode; iii) co-expression of multiple markers from distinct cell lineages (e.g. *PTPRC, FLT1, MME, SLC12A1*). Inversely, all barcodes labelled as doublet were separately clustered and previously misclassified doublets were retained.

After initial doublet identification, the full GEX and ATAC datasets were loaded, and doublets or barcodes not assigned to a donor were removed. Further, any gene expressed by less than 20 nuclei barcodes was removed. Additionally, barcodes with i) unique molecular identifiers (UMIs) < 300 or > 12,000; ii) unique genes > 4000; iii) mitochondrial gene content > 10%; iv) number of ATAC fragments < 700 or > 100,000; v) number of unique peaks < 300 or > 40,000; vi) a nucleosomal signal score (calculated with NuceosomeSignal) < 2; vii) a TSS enrichment score (calculated with TSSEnrichment) < 2 were removed (Supplementary Figs. S2d-g and S3a, d).

Next, dimensionality reductions were independently computed based on GEX and ATAC datasets. In both cases datasets were split by the donor sex using SplitObject. Data was normalised using SCTransform with option vst.flavor = "v2" as previously and integration anchors were computed using FindIntegrationAnchors with the top 3000 shared variable features. The data was integrated using IntegrateData with dims = 1:80 and the UMAP embedding was computed with dims = 1:80. Similarly, the ATAC data was processed using FindTopFeatures, RunTFIDF, RunSVD with $n = 100$ before splitting by donor sex, computing anchors using all peaks, and integrating data with IntegrateData with dims = 1:80 and RunUMAP with dims = 2:80). Finally, the Seurat weighted nearest neighbour (WNN) analysis was used to compute a joint dimensionality reduction using FindMultiModalNeighbors with options dims.list = list(1:100, 2:100) and k.nn = 60) in the PCA and LSI embeddings of the GEX and ATAC datasets respectively. The joint UMAP was computed using the WNN graph with RunUMAP option nn.name = "weighted.nn" and barcodes were clustered using FindClusters with options resolution = 4 and graph.name = "wsnn".

In order to refine the peak set and detect peaks present in small cell populations clusters were tentatively annotated (pending the final annotations) and the Signac CallPeaks wrapper function for MACS2 (v2.2.7.1)[51] was used to call cluster specific peaks. Peak sets for each cluster were further merged and pruned by removed peaks in scaffold and blacklisted regions. Finally, a peak by nuclei barcode count matrix was generated using the FeatureMatrix function with the new peaks regions as features. Following this, ATAC and WNN dimensionality reductions were recomputed as before, and barcodes were re-clustered (Supplementary Fig. S3b). To determine final cluster annotations barcodes were first broadly clustered using the FindClusters function with lower resolution settings. Broad cluster (level 1 annotations) were further iteratively subclustered by re-computing dimensionality reductions and sub-clusters within a larger cluster. Marker genes for each cluster were determined using the FindMarkers function and remaining subclusters co-expressing signatures of distinct cell types were removed as doublets. Each subcluster was manually annotated (level 2 annotations) according to gene markers determined by prior human kidney atlases[4] (Supplementary Fig. S6, Supplementary Data 4). We defined injured epithelial clusters based on a proportional enrichment in UUO samples compared to control samples as well increased expression of generic injury genes such as *PROM, DCDC2, SPP1, ITGB6, ITGB8* (Fig.1c−e, Supplementary Data 5, 8). Subclusters of PT and LOH nephron segments were identified as inflammatory cell states based on the expression of injury markers as well expression of chemokines, extracellular matrix remodelling and

adhesion factors and markers of cell cycle arrest (Fig. 2c). Individual chemokines were identified in other injured nephron segments, however they lacked a consistent signature of multiple inflammatory markers and sufficient separation from injured counterparts (Supplementary Fig. S12). Agreement between GEX and chromatin profile of markers genes was assessed using the CoveragePlot function in genomic loci of marker genes determined by GEX (Supplementary Figs. S4, S5).

All further basic graphs showing gene expression, chromatin state, and meta data features were generated using Seurat and Signac functions DimPlot, FeaturePlot, DotPlot, VlnPlot and CoveragePlot.

## Integration with KPMP snCv3 dataset

The full (as of September 2023) The Kidney Precision Medicine Project (KPMP) snRNA-seq dataset along with associated clinical data were downloaded from the KPMP website (https://atlas.kpmp.org). Integration of the snCv3 dataset data was performed using Seurat with the multiome dataset as reference. Transfer anchors were computed with FindTransferAnchors using dims = 1:50 with the multiome precomputed GEX principal components as reference. Subsequently, the snCv3 data along with prediction of respective multiome level 1 and level 2 cell annotations were projected on the WNN UMAP reduction using the MapQuery function (Supplementary Figs. S6a, 9). Agreement of original snCv3 annotations multiome annotations were assessed using sankeyNetwork graphs implemented by the networkD3 package (v0.4) (Supplementary Fig. S9b).

## Differential abundance analysis

Abundance of different cell types in samples was calculated as proportion of total cells or proportion of total epithelia cells (non-Immune/Endothelial/Interstitial cells) to normalise for shifts in overall composition due to immune cell influx in UUO samples. Comparisons between cell populations in control and UUO samples were performed using a Wilcoxon signed-rank test using the R function wilcox.test with the option alternative = "two.sided" (Figs. 1e and 2b, Supplementary Fig. S12f, i).

## RNA velocity and pseudotime analysis

Count matrices of spliced and unspliced reads were generated from cellranger GEX BAM files for each library using velocyto (v0.17.15)[52]. The gene model from the 10X hg38 reference genome along with a mask for repetitive regions downloaded from the UCSC genome browser were passed to the velocyto run command along with BAM files and a whitelist of nuclei barcodes retained in the final analysis. RNA velocity streams were inferred independently for PT, other epithelial and myeloid cell subclusters using scVelo (v0.2.5)[53]. Velocyto loom files were imported using the scv.read function and UMAP embeddings and cell annotations were imported to the AnnData object. The data was normalised according to the standard workflow and moments across 50 nearest neighbours in the first 50 principal components were computed. Splice kinetics were then modelled using a likelihood based dynamical models and visualised using pl.velocity_embedding_stream (Fig. 2a, Supplementary Fig. S12c, g).

To infer pseudotime orderings for PT and LOH subsets we used the slingshot R (v2.8.0) package[54]. For PT trajectories we considered healthy PT S1−S3 as start points and PT Inflammatory cells as end point. Similarly, for LOH trajectories cTAL1, cTAL2 and mTAL clusters were considered as starting point and LOH Inflammatory as end point, while ignoring Macula Densa cells. The getLineages with relevant start and end clusters was used to construct the minimum spanning tree and getCurves was used to get a smoothed ordering of cells along the pseudotime trajectory. Gene expression and chromatin accessibility dynamics correlated with the inferred pseudotime values were identified using the tradeseq package, using the fitGAM function to fit regression models to counts and pseudotime values. Normalised

count values of correlated genes or peaks were then extracted from the Seurat object and smoothed using the smoothing function (with strength = 0.2) from the modelbased R package (v0.8.6.3). Heatmap plots of smoothed dynamics were then visualised using the pheatmap package (v1.0.12) (Figs. 2g and 7b, Supplementary Fig. S12d).

## Gene set analysis

We defined differentially expressed genes between PT Injured/Inflammatory and PT S1-S3 cells using the FindMarkers Seurat function. Up- or downregulated genes with an absolute average log2-fold change > 0.25 and adjusted $p$-value < 0.05 were forwarded for over-representation analysis implemented by the clusterProfiler R package (v4.10.0)[55]. Using the enrichGO function enrichment for GO biological process terms with < 600 genes were tested. To project the most significantly enriched non-redundant GO terms onto the pseudotime trajectory, genes associated with each GO term were extracted and cells were scored using the AddModuleScore_UCell function. A locally estimated scatterplot smoothing curve was fitted to scaled GO scores as implemented by geom_smooth with a span smoothing strength of 0.4 (Fig. 2h).

## Gene regulatory network analysis

We used the SCENIC+ python (v1.0.0) package[56] for gene regulatory network inference and TF analysis. For this, a custom cistarget database overlapping with ATAC peaks found in the dataset was created by generating a FASTQ file containing peak regions using the bedtools getfasta command. The create_cisTarget_databases.py script was then used with the SCENIC+ motif collection (provided by SCENIC+) to score motif sequences in each ATAC peak.

Next, the Seurat dataset was subsampled to <=1000 cells per level 2 annotation and exported to python (v3.8.0). The remaining SCENIC+ workflow was followed with minor alterations. Briefly, GEX and ATAC were pre-processed with default settings, chromatin topic modelling was performed using pycisTopic, and 70 topics were selected for the dataset. PycisTarget was then used to find motif enrichments in accessible chromatin regions, enhancer to gene relationships and TF to gene relationships were calculated and the gene regulatory network was constructed using SCENIC+ functions.

The derived regulons consisting of TF-CRE-gene links were further pruned by removing regulons with negative CRE to gene correlations and regulons with <15 target genes. In total we detected filtered regulons for 474 unique TF motifs found in 109,147 unique regions linked to 11,567 genes. The filtered regulons were exported to R and cells were independently scored by calculating scores for TF gene targets and CRE targets of each TF using the AddModuleScore_UCell function. A combined TF score for each TF in each cell was calculated as mean of the gene target and CRE target scores. For all further analysis of gene regulatory networks in specific cell types (e.g. PT cells) only genes and CRE expressed or accessible in the given cell type were considered to generate cell-type specific regulons. Genes were considered expressed in the cluster if i) for genes with global expression level below expression by 10% of cells the gene is expressed by at least 2% of cells in a subcluster ii) for genes with a global expression level above 10% it is expressed by an equal or higher proportion of cells than the global expression level iii) it is expressed by more than 20% of cells. Similarly, regions were considered accessible if the same thresholds were met. Other regions and genes were considered not expressed and removed from the analysis. For linkage analysis between TFs and target genes the linkages between TFs, CREs and target genes were derived from SCENIC+ regulons (Figs. 6 and 7, Supplementary Data 22). Overlap between TF target genes was assessed by calculating the Jaccard index between lists of target genes of individual TFs (Fig. 6d). TF score dynamics along pseudotime trajectories were visualised by fitting a locally estimated scatterplot smoothing curve with a smoothing factor of 0.4 using the geom_smooth function (Fig. 6f).

To determine how our predicted HNF4A target genes compared with those determined by experimental validation, we utilised published bulk RNA-seq data from human kidney organoids with and without *HNF4A* knockout[30]. Log-fold changes in *HNF4A* KO versus WT organoids were computed using DESeq2. Genes were ordered by log-fold change with higher values representing greater expression in WT organoids. This gene list was compared with HNF4A target genes (derived from SCENIC+ regulons described above) as well as genes upregulated (LFC > 0.1 and adjusted $p$-value < 0.05) in healthy PT cells (S1-S3) versus injured and inflammatory PT cells from our snRNA-seq analysis using the GSEA function (ClusterProfiler v4.10.0).

## Ligand-receptor analysis

Analysis of ligand-receptor interactions was performed using the CellChat (v2.1.0) package[57]. We asked how different PT subclusters interact with immune cells and fibroblast, therefore the dataset was subset to these cell types. To infer interaction the CellChat workflow was followed with default setting, except adjusting the truncated mean gene filtering to 0.01 in the computeCommunProb function in order to account for low expression levels observed particularly by chemokine genes. For visualisation purposes ligands and receptors overexpressed in injured or inflammatory PT cells were extracted and plotted using the Seurat DotPlot function for the ligands and receptors identified by CellChat. Interactions were manually connected using the CellChat interaction database as reference, while omitting individual receptors with no expression or non-relevant expression patterns due space constrains (Fig. 5a, h, k).

## Re-analysis of mouse sc/snRNA-seq datasets

Count matrices from previously published datasets were downloaded from gene expression omnibus under accession numbers GSE140023[15] and GSE139107[9]. In both cases the count matrices were loaded in R and cells were subset to clusters identified as PT cells in the original publication. For both datasets Seurat RPCA integration was used to account for batch effects in experimental design. For the R-UUO datasets the data was split by the timepoint of sample collection (Control, UUO2, UUO7, R-UUO) and each subset was independently normalised using SCTransform with the option vst.flavor = "v2" and principal components were computed using RunPCA with npcs = 100. Integration anchors were computed using FindIntegrationAnchors with options reduction = "rpca" and k.anchor = 30 and subsets were integrated using IntegrateData with dims = 1:50. Subsequently the UMAP dimensionality reduction and clusters were computed using RunUMAP, FindNeighbors and FindClusters with the first 50 principal components. Analogously, the IRI data integrated using the replicate information provided in the GEO records.

In both cases cluster markers were computed using the FindAllMarkers function with options min.pct = 0.01 and logfc.threshold = 0.1. Similarly, we found clusters expressing injury markers such as *Havcr1* and *Vcam1* were enriched post-injury. We identified subclusters expressing markers of maladaptive repair[9], showing similar expression to inflammatory PT cells in human kidneys such as *Ccl2*, *Cxcl1*, *Icam1* and *Tgfb2*. Notably, the expression of *Vcam1* and *Dcdc2a* was more restricted to this cluster and not found in generic injury clusters in both datasets, as opposed to humans where respective *VCAM1* and *DCDC2* genes were more widely expressed (Supplementary Fig. S10d). Due to overall similarities, we followed naming conventions using for the human data. Finally, the proportion of each cell state was calculated as the proportion of total PT cells in each animal model at each timepoint and visualised as time series graph (Fig. 2i).

## CosMx spatial molecular imaging

**CosMx 6000-plex profiling of UUO and control nephrectomy specimens.** A subset of samples used for multiome snRNA/ATAC-seq was used to perform imaging-based spatial transcriptomics using the

CosMx SMI platform, with a total of five UUO and two control samples being profiled (Supplementary Data 9). Microscope slides were prepared by cutting 5 µm sections from FFPE embedded tissue blocks and adhered to Superfrost Plus slides (VWR, 631-0108). All sections were positioned in the pre-defined capture area (defined by the CosMx flow cell dimensions). Following section placement sample preparation was performed using the Human RNA 6k Discovery kit (NanoString).

## CosMx 1000-plex profiling of CKD specimens

Nine needle biopsy samples and four nephrectomy samples were profiled using the 1000-plex workflow (Supplementary Data 13). 5 µm sections were cut from blocks and three biopsy or four nephrectomy samples were arranged in the capture area. In total, four slides were profiled using the Human RNA Universal Cell Characterization kit (NanoString).

## Sample processing

Prepared slides were processed according to the manufacturer instruction (either 6000-plex or 1000-plex RNA kit). Briefly, slides were deparaffinized with two 5 min Xylene washes and two 2 min 100% ethanol washes followed by proteinase K (3 µg/ml; Thermo Fisher, AM2546) digestion (40 °C for 30 min) and heat induced epitope retrieval at 100 °C for 15 min in ER1 buffer (Leica, AR9961). After rinsing with nuclease free water, slides were incubated with 1:1000 diluted fiducials (Bangs Laboratory) in 2X SSCT (2X saline sodium citrate, 0.001% Tween 20) for 5 min at room temperature. After PBS washes, slides were fixed with 10% neutral buffered formalin for 5 min at room temperature. After rinsing with Tris-glycine buffer (0.1 M glycine, 0.1 M Tris-base) and a 5 min wash with PBS, samples were blocked using 100 mM N-succinimidyl acetate (Thermo Fisher, 464750250) in NHS-acetate buffer (0.1 M NaP, 0.1% Tween pH 8) for 15 min at room temperature. Samples were rinsed with 2X saline sodium citrate (SSC) for 5 min and covered with an Adhesive SecureSeal Hybridization Chamber (Grace Bio-Labs).

The probe panel[58] was denatured at 95 °C for 2 min and placing on ice. The ISH probe mix was prepared using 1 nM ISH probes in 1X Buffer R and 0.1 U/µL SUPERaseIn RNase Inhibitor (Thermo Fisher, AM2696). Samples were then incubated with the probe mix at 37 °C overnight. On the next day slides were washed with 50% formamide (VWR) in 2X SSC at 37 °C for 25 min and rinsed twice with 2X SSC for 2 min at room temperature. Before antibody incubation slides were treated with blocking buffer containing DAPI nuclear stain for 15 min. For visualization of cell morphology antibodies against C298/B2M, PanCK, CD45 and CD3 from the CosMx Segmentation and Supplemental Markers Kit were chosen and incubated at room temperature for 1 hours. Finally, after washing with 2X SSC for 2 min at room temperature custom-made slide covers were attached to the sample slide to form a flow cell.

## Image acquisition

RNA readouts were acquired according to previously described protocols[58]. Briefly, the assembled flow cells were loaded onto the SMI instrument and washed with reporter buffer. The flow cell was washed and a median of 12 and total of 140 field of views (FOVs) were defined inside which RNA was detected. The imaging procedure was repeated with 16 reporter pools, with each imaging step repeated 8 times to increase RNA detection sensitivity. Each cycle was initiated by flowing 100 µL of the Reporter Pool into the flow cell and incubating for 15 min followed by flushing with 1 ml of Reporter Wash Buffer. The wash buffer was replaced with imaging buffer and nine Z-stack images (0.8 µm step size) of each FOV were acquired. Each cycle was completed with UV cleavage of fluorophores on the reporter probes and washing using Strip Wash buffer.

Following RNA readouts, staining for DAPI and the four morphology antibodies was acquired by capturing nine Z-stack images after washing with Reporter Washing Buffer.

## Image processing and cell segmentation

Image processing and feature extraction followed the NanoString pipeline previously described[58] which extracted XYZ locations of individual transcripts. After image registration, feature detection and localisation, the Z-stack images the nuclear and cell membrane markers were used to define cell boundaries using a machine learning algorithm[58]. Cell segmentation boundaries were used to assign transcripts to cells from which a count matrix was generated. Pipeline outputs included a cell by gene matrix, cell segmentation polygons, and cell meta data.

## 6,000-plex dataset processing, cluster identification and dimensionality reduction

Pipeline outputs were imported into Seurat using the LoadNanostring() function. Initial quality control included assessment of the number of transcript and unique genes detected in individual cells and in a spatial context (e.g. lower quality fields of view). As no major spatial effects were observed all cells with >20 transcripts were retained for further analysis. Transcript count data was normalised using SCTransform with option vst.flavor = "v2".

To determine broad cell type identities a combination of clustering and segmentation was used. Firstly, immunofluorescence images (included in CosMx pipeline output) were loaded in QuPath (v0.5.0) to manually segment glomeruli (based on morphology) using polygon annotations. Cells in the Seurat object falling within glomeruli polygon boundaries were extracted and assessed separately. Dimensionality was reduced using RunPCA with option npcs = 100, samples were integrated using RunHarmony and UMAP coordinates and clusters were computed with 30 principal components using RunUMAP, FindNeighbors and FindClusters respectively. Clusters were annotated based on marker gene expression.

Next, non-PT tubular segments staining positive for pan cytokeratin (PanCK) and immune cell subsets expressing CD45 and/or CD68 on a protein level were identified. For this mean fluorescence intensities per cell (included in CosMx pipeline output) were log-10 transformed and appended to the RNA count matrix. UMAP embeddings and clusters were computed as described above. PanCK positive clusters were extracted and re-clustered and clusters corresponding to loop of Henle, distal convoluted and connecting tubules as well as principal and intercalated cells were identified based on distinct RNA expression and differences in PanCK intensity (highest in collecting duct segments, lowest in loop of Henle). Lastly, any remaining cells mainly consisting of proximal tubules, endothelial and mesenchymal cells were re-clustered with the same parameters.

After broad cell types were annotated finer sub-clusters were defined individually for each broad cluster. Lymphoid and myeloid cell subclusters were defined by projecting cells onto respective clusters in the multiome snRNA/ATAC-seq dataset presented here. For this the RNA modality was subset and a new temporary embedding was computed using RunUMAP with the first 40 principal components. The spatial dataset was projected onto this embedding using FindTransferAnchors dims = 1:40, k.anchor = 20, k.score = 80) and MapQuery. Similarly, interstitial clusters were defined by projecting the spatial dataset onto mesenchymal clusters in the multiome dataset with the same settings. Epithelial injury states were defined by individually sub-clustering each nephron segment. For each segment the UMAP embedding and cluster were computed using RunPCA (npcs = 100), RunHarmony, RunUMAP (30 dimensions), FindNeighbors (30 dimensions) and FindClusters. Injury states were defined in agreement with the clusters identified in the multiome dataset, with sub-clusters expressing injury markers (such as *VCAM1*, *TPM1*, *VIM*, *SPP1*, *ITGB6*, *ITGB8*) defined as injured cells and clusters additionally enriched with inflammatory markers (such as *CCL2*, *CXCL1* and *MMP7*) defined as inflammatory cell states (Supplementary Figs. S7 and 8).

Lastly, a UMAP embedding for the complete spatial dataset was computed by projection onto the multiome dataset. The multiome embedding was computed as RNA-based UMAP using RunUMAP (40 dimensions). Spatial data was projected using FindTransferAnchors (dims = 1:40, k.anchor = 30, k.score = 80) and MapQuery.

## 1,000-plex dataset processing, cluster identification and dimensionality reduction

Data was initially imported into Seurat using the LoadNanostring() function. The number of transcripts and genes per cell were assessed for regional biases (Supplementary Fig. S14a, b) and cells with < 20 transcripts were excluded from quantitative analysis but not from image plots. Batch effects between biopsy and nephrectomy samples were observed and therefore integrated using the harmony (v0.1) package[59].

For this, each group was independently normalised with SCTransform using vst.flavor = "v2", and principal components were computed with RunPCA with option npcs = 100. RunHarmony was then used to align principal components and the UMAP projection was computed on adjusted principal components. Cell types were defined in an iterative manner similarly to the multiome dataset by first defining broad clusters resembling distinct cell lineages, followed by sub-clustering individual clusters.

Cell type annotations were guided by marker genes previously identified in the multiome dataset. As the probe panel did not target kidney-specific genes similar clustering resolutions were not possible, however we could resolve broad nephron segment signatures based on distinct marker expression patterns. For example, *EGF* expression characteristic of LOH and DCT tubule cells was observed, but due to lack of additional markers discriminating these two groups robust subclusters could not be derived. Therefore, epithelial cells of the nephron were broadly divided into PT, LOH-DCT as well as collecting duct (CD, with resolved PC and IC) segments. Notably, no CNT and ATL/DTL markers were detected, and these are likely grouped together with other segments. Using the previously defined injury and inflammatory markers we next searched for patterns of epithelial cell activation by sub clustering the broader epithelial clusters. We observed expression of injury markers such *VCAM1*, *TPM1*, *VIM*, *SPP1*, *ITGB6* and ITGB8. Further, markers predominantly expressed by inflammatory cells such as *CCL2*, *CXCL1*, *ICAM1* and *MMP7* were co-expressed by subsets of epithelia, confirming expression patterns observed in the multiome dataset (Fig. 4f). Therefore, these cells were annotated following these conventions. We observed expression patterns clearly identifying other non-epithelial cell types such as endothelia, fibroblasts, smooth muscle cells, mesangial cells, myeloid cells as well as T and B lymphocytes. These were further subclustered if discrete populations were observed, e.g. myofibroblasts are distinguished by expression of *COL1A1* and *COL3A1* in addition to shared fibroblast markers (Supplementary Fig. S15). In addition to level 1 and level 2 annotation, for visualisation purposes we derived a further annotation summarizing healthy, injured, and inflammatory epithelia as well as broad non-epithelial cell types (Supplementary Fig. S14d). Visualisations of cell segmentations and transcript locations were generated using Seurat's ImageDimPlot function. For epithelial cell clusters proportions were calculated as the number of cells relative to the total of that cell type, while for non-epithelial cell proportions were calculated relative to the total number of cells.

## Niche analysis

For both spatial datasets, to define spatial niches a cell x neighbouring cells matrix was constructed by counting neighbouring cell types within a 75 μm radius. To account for differences in cell density, each entry was subsequently divided by the number of observed neighbours. Niches were defined using the Mclust algorithm with an EII model. Initially, 20 clusters were defined and annotated according to the most prominent cell type or anatomical structure in the cluster. Similar clusters were collapsed niches with coherent themes were defined (Figs. 4e and 5b). Enrichment of cell types within niches was assessed by calculating a log2-fold enrichment ratio defined as the quotient between the observed and expected number of cells of cells (proportions of the cell type across the dataset) plus 1. After log-transformation the data was centred on 0 (equivalent to no enrichment) by subtracting 1 (Supplementary Fig. S16c). Correlation between the proportion of cells in the fibrotic niche and eGFR or %fibrosis by PSR staining was assessed using a linear regression slope with Pearson correlation coefficient and associated *p*-value (Supplementary Fig. S16d).

## Cell neighbourhood analysis

We inferred enrichment of a given cell type ($ct_{query}$) in proximity to other cell types by first defining a search radius of 25 μm measured from the centroid (of each cell's boundary polygon) to the centroid of a neighbouring cell. Next, we generated a $ct_{query}$ by $ct_{neighbour}$ count matrix by counting the number of encounters of each level 2 annotations within the search radii of each query cells. Observed counts were then normalised by the absolute number of cells of the respective cell type encountered in the tissue section giving the observed frequency $f_{obs}$.

Simultaneously, we calculated the expected frequency $f_{exp}$ as the proportion a given cell type across the entire tissue sample multiplied by the total number of cells encountered within the search radius (equivalent to random tissue architecture). The enrichment ratio was then defined as the log2-fold ratio between ($f_{obs}$ / $f_{exp}$) + 1 and calculated for each $ct_{query}$ to $ct_{neighbour}$ pair.

To derive a symmetric matrix the log2-fold enrichment ratios of a given $ct_{query}$ to $ct_{neighbour}$ pair were multiplied with each other. For visualisation purposes enrichment values were log2 transformed and centred on 0 by substracting 1 (with > 0 indicating positive enrichment). We calculated the enrichment ratio for each tissue sample individually and used the mean across all samples for visualisation as heatmap (Fig. 3b, Supplementary Fig. S17d). For individual pairs we assessed statistical significance of enrichments across samples using a paired two-sample Student's *t* test (Figs. 3c and 4h).

## Transcript spatial enrichment

In order to assess transcript enrichment in proximity of a given cell type we defined search radii up to a distance of 50 μm in 1 μm steps from the cell centroid. For a given cell type the number of detected transcripts of a given species in each interval were then counted and normalised by the area of the circle segment and the number cells in the area. Normalised transcript counts were averaged over all tissue sections and plotted as log2 + 1-fold enrichment over a sample of 10,000 random cells (independent of cell type) (Figs. 3d and 4g, Supplementary Fig. S17e).

## Spatial density maps

Density maps were used to visualise the spatial distribution of cell types or individual transcripts. In both cases co-ordinates were extracted from the seurat object and plotted in 2D space. 2D kernel density estimation was used to construct density maps using the kde2d function (MASS, R package) and visualised as contour bands. For cell type density maps a bandwidth of 0.1 and for transcript a bandwidth of 0.8 was chosen.

## Cell type composition of transcript origins

To compute the cell type origins of individual transcripts within the fibrotic niche the spatial dataset was subset to only include cells located within the fibrotic niche. Next, the sum of transcripts counts was computed for each cell types. To account for noise (randomly distributed false transcript detections) observed within the CosMx

assay, 0.09 was subtracted from the transcript sum for each cell. Finally, the adjusted sum of transcript per cell type was represented as proportions to the total adjusted counts of the transcripts within the fibrotic niche.

## Cell culture

Primary renal proximal tubule epithelial cells (RPTEC/TERT1, ATCC CRL-4031) were cultured at 37 °C in a 5% CO2 incubator using custom medium containing 25 ng/ml hydrocortisone (Merck, H0888), 3.5 μg/ml ascorbic acid (Sigma-Aldrich, A4403), 1% insulin-transferrin-selenium-ethanolamine (ITS-X) (Thermo-Fisher Scientific, 51500056), 6 pM Triiodo-L-Thyronine (Merck, T6397), 25 ng/ml Prostaglandin E1 (Merck, P8908), 0.8% v/v of 0.28 g/ml N-2-hydroxyethylpiperazine-N-2-ethane sulfonic acid (HEPES, Sigma-Aldrich, H4034) dissolved in 1 M sodium hydroxide, 10 ng/ml recombinant human Epidermal Growth Factor (Promega, G5021) and 0.1 mg/ml geneticin (Gibco, 10131027) in Dulbecco's Modified Eagle Medium (GIBCO, 31331093). All experiments were performed with cells between passage 3 and 6.

For irradiation experiments, cells were plated in 6 well plates (200,000 cells in 2 ml media) then administered 10 Gy of irradiation 3 days later using a Caesium-137 gamma irradiator (CIS Bio International Gamma Irradiator type IBL 637). Control cells were 'mock' irradiated (taken to irradiator but not exposed to irradiation). Media was changed immediately following (+/−mock) irradiation and again 3 days later. SA-β-GAL staining (Cell Signalling Technology) was performed as per the manufacturer's instructions and imaged on the EVOS FL Auto 2 microscope. Images were analysed in in Image J (version 8). Separate macros were run for total cell counting and SA-β-GAL positive cell counting.

Samples were collected for RNA analysis 7 days after irradiation using 1 ml of TRIzol (Invitrogen, 15596026). The suspension was stored at −80 °C until RNA extraction. RNA was extracted using the RNeasy Plus micro kit (Qiagen) and included the optional DNase step. The RNA was eluted in 14 μl RNase-free water and concentration estimated using a Nanodrop One (Thermo- Fisher Scientific). RNA quality was assessed using the RNA 6000 Pico Kit (Agilent, 5067-1513) for RNA integrity numbers (RIN) > 7. Extracted RNA was sent to Genewiz where Poly-A enriched libraries were prepared using the NEBNext Ultra II RNA Library Prep Kit for Illumina following manufacturer's instructions (New England Biolabs, E7760).

For CUT&RUN experiments were plated at 0.8 ×10⁶ cells/well into 6-well plates. The next day cells were treated with recombinant human TNF (20 ng/ml, Bio-Techne, 210-TA-020). After 24 h cells were detached using 0.05% Trypsin and 0.02% EDTA in PBS for 8 min and resuspended in culture medium before nuclei isolation.

## Bulk RNA-seq analysis

RNA-seq libraries from RPTEC irradiation experiments were sequenced on an NovaSeq 6000 platform (Illumina) with a mean output of 159 million 150 bp paired-end reads per sample. After sample demultiplexing, adapter sequences were trimmed from FASTQ files using Trim-Galore (Cutadapt v2.8). Reads were aligned the GRCh38 (hg38) reference genome (GENCODE primary assembly) with a mean alignment rate of 84.3% using STAR (v2.7.9a)[60] with the option --quantMode GeneCounts. Gene counts were imported to R and analysed using DESeq2 (v3.18)[61]. Genes with less than 200 overall counts were removed from the analysis. Differential expression analysis was performed using the DESeq function with default settings while controlling for variations due to replicate number in the model design. For visualisation purposes the count data were normalised using a variance stabilising transformation implemented by the DESeq2 vst function and scaled before plotting.

## Re-analysis of ABT-263 RNA-seq data

Count matrices of previously published RNA-seq data were downloaded from gene expression omnibus under accession numbers GSE157866[38]. Transcript abundances (Salmon transcript-level quantification) were imported into R and summarized to a gene level using the tximport function. Gene IDs were subsequently annotated to MGI nomenclature using BioMart functionality and the data was analysed using DESeq2 as described previously. Briefly, a DESeq2 object was created using DESeqDataSetFromMatrix with treatment and replicate as the design formula. Genes with less than 10 gene total gene counts were removed and the data was normalised using the DESeq and vst function with default setting. After Variance stabilizing transformation expression and gene-wise scaling, expression values of previously identified markers were visualised using pheatmap. Sample 4 from the 'vehicle post R-UUO' group was excluded from visualisation, due to behaving during outlier in principal component analysis and gene expression not behaving similarly to other samples in the group, leaving 5 remaining samples in the group.

To deconvolute bulk transcriptomes and estimate proportions of PT cell states the MuSiC R package (v1.0.0) was used[62]. Reference transcriptomes were derived from mouse snRNA-seq data[9]. Deconvoluted proportions were calculated using the music_prop function, with select.ct including healthy, injured and inflammatory cell states in the snRNA-seq dataset and markers including the top 2000 differentially expressed genes between PT cell states. Results were compared using a Student's $t$ test.

## CUT & RUN

**Library preparation.** CUT&RUN libraries from TNF-treated RPTECs were generated using the CUTANA ChIC/CUT&RUN kit (EpiCypher, 14–1048) with provided reagents unless indicated. After cell harvest samples nuclei were isolated. For this, cells were resuspended in 500 μl cold nuclei extraction buffer containing 2 mM HEPES pH 7.9, 10 mM KCl, 0.1% Triton X-100 (Merck, X100), 20% glycerol, 1X cOmplete protease inhibitor cocktail (Roche, 11873580001), 0.5 mM spermidine in PBS. After incubation on ice for 5 min, nuclei were centrifuged (500 × $g$, 5 min, 4 °C) and the nuclei pellet was resuspended in nuclei extraction buffer without Triton X-100. Nuclei were counted using a LUNA-FX7 cell counter (Logos Biosystems) and 0.5 × 10⁶ nuclei were used as input for the CUT&RUN assay.

The CUT&RUN assay was then performed following manufacturer instructions. Briefly, activated Concanavalin A (ConA)-conjugated paramagnetic beads were bound to nuclei. Using a magnetic rack bead-bound nuclei were washed and resuspended in antibody buffer. For control reactions 0.5 μg IgG negative control antibody (EpiCypher, 13-0042) was used while 0.5 μg of NFKB1 (Cell Signalling Technology, 13586S) or JUN (EpiCypher, 13-2019) were added for positive reactions. After overnight incubation and washes pAG-MNase was added and chromatin was digested by addition of $CaCl_2$ solution. Digestion was stopped by addition of the Stop Master Mix and the supernatant was transferred to a DNA clean-up column. After washes DNA was eluted in 12 μl of DNA Elution Buffer and quantified using the Qubit dsDNA Quantification kit (Thermo Fisher Scientific, Q32851).

For each reaction 5 ng purified DNA were used as input for library preparation using the CUTANA CUT&RUN Library Prep Kit (EpiCypher, 14-100) according to manufacturer instructions. Library was quantified using the Qubit dsDNA Quantification kit and fragment size was assessed using the Bioanalyzer High Sensitivity DNA Analysis kit (Agilent, 5067-4626).

## CUT&RUN analysis

Libraries were sequenced on a NextSeq 2000 platform (Illumina) with 100 bp paired-end reads and a mean read depth of 10 million reads per library. After library demultiplexing remaining Illumina indices were removed using Trim Galore (Cutadapt v2.8) and aligned to the GRCh38 (hg38) reference genome with bowtie2 --local --very-sensitive --no-mixed --no-discordant --phred33 -I 10 -X 700 (v2.5.2)[63]. Peak calling was

performed using MACS2 (v2.2.7.1) with default settings. The peak significance threshold was determined using p-values derived from MACS2 with manual inspection of peak calling sensitivity at expected locations. A threshold of -log10(p-value) of 5 and 4 were used for JUN and NFKB1 peaks respectively (Supplementary Fig. S21). The BAM files were converted to BigWig format using the bamCoverage command from DeepTools (v3.5.4) with the normalisation option --normalizeUsing CPM and visualised using Signac's CoveragePlot() function. Overlap of ATAC peaks with CUT&RUN peaks was calculated requiring a 10% overlap in both peaks using the bedtools (v2.31.0) intersect function with parameters -wao -r -f 0.1. The overlapping ATAC peaks were used to infer differential accessibility in these regions using the RunPresto function with parameters logfc.threshold = 0, min.pct = 0.00 or find motifs using the Signac function FindMotifs with default settings (Fig. 7f). The fraction of ATAC reads in CUT&RUN peaks for each cell was calculated using Signac's FeatureMatrix() function using the original CUT&RUN peaks in BED format as the feature set to be quantified (Supplementary Fig. S21).

## Statistics

Comparisons of clinical and histological characteristics between the unobstructed and obstructed groups were performed by two-sided Student's *t* test or Chi-test for continuous or categorical data respectively. In the CosMx analysis, the relationship between abundance of cell types and eGFR or percentage fibrosis area was assessed using a linear regression slope with Pearson correlation. In the AP-1 inhibitor experiment, comparisons between the vehicle and T5224 groups were assessed by Student's *t* test.

All sample sizes, measures of data dispersion and statistical tests are provided in the relevant figure legends

## Reporting summary

Further information on research design is available in the Nature Portfolio Reporting Summary linked to this article.

## Data availability

The raw and processed data generated in this study have been deposited on NCBI Gene Expression Omninbus (GEO). Multiome snRNA/snATAC-seq data are available under the accession GSE254185. CosMx 6,000-plex pipeline outputs and processed data are available under accession GSE282059. CosMx 1,000-plex pipeline outputs and processed data are available under accession GSE253439. Bulk RNA-seq data of irradiated RPTECs and CUT&RUN data can be found under accessions GSE252584 and GSE254187 respectively. Multiome snRNA-seq data can be visualised interactively on CELLxGENE [https://cellxgene.cziscience.com/e/867757c1-3b1a-49d9-a0cd-17767eb160cc.cxg]. Additional data are available on Zenodo under the accession code 15124887. Data sets for murine R-UUO and IRI models are available on GEO under accessions GSE140023 and GSE139107 respectively. Source data are provided with this paper.

## Code availability

Scripts to reproduce figures are available on github [https://github.com/mreck1/human_kidney_multiomics] and on Zenodo under the accession code 15124887.

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

## Acknowledgements

This work was funded by Kidney Research UK grants (RP_024_20190306 and RP_013_20221129) to B.C. A PhD studentship from the MRC Precision Medicine Programme, grant number MR/N013166/1 to M.R. An MRC Clinical Research Training Fellowship (MR/W00089X/1) to D.B. A pre-doctoral fellowship from the American Heart Association (967503) and a PhD studentship from University of Edinburgh College of Medicine and Veterinary Medicine Scholarship to S.V. A Kidney Research UK Senior Fellowship (SF_001_20181122) to L.D. An MRC (grant number MR/W030322/1) and a Career Fellowship Award from the Scottish Chief Scientist Office to A.L. An MRC Senior Fellowship (MR/X006735/1) to DF. We are grateful for the assistance of the flow cytometry core facility, University of Edinburgh for their assistance with nuclei isolation and to Melanie McMillan for assistance in cutting biopsy sections. We are also grateful to Mr Steve Leung and Mr Edward Mains, Consultant Urologists; Dr Marie O'Donnell, Consultant Uro-pathologist, Western General Hospital, Edinburgh and NHS Lothian Bioresource for assistance with sample selection and acquisition.

## Author contributions

Conceptualisation: M.R., J.H., T.C., D.F., B.C.; data generation and curation: M.R., D.B., S.V., C.S., R.M.B.B., H.H., C.C., P.J., R.C., A.N., W.Y., N.S., C.W., E.O.S., M.B., A.C., L.D.; analysis: M.R., D.B., S.V., C.B.; funding acquisition: C.B., T.C., D.F., B.C.; sample acquisition: M.R., C.B., A.L., D.F., B.C.; writing, reviewing and editing the manuscript: all authors. M.R. and D.B. contributed equally to the work. T.C., D.A.F. and B.C. contributed equally to the work.

## Competing interests

A.N., W.Y., N.S., C.W. are current or previous employees and shareholders of NanoString Technologies, now Bruker Spatial Biology. The remaining authors declare no competing interests.
