## [Transparent Peer Review file · Nature Communications]

Multiomic analysis of human kidney disease identifies a tractable inflammatory and pro-fibrotic tubular cell phenotype

Corresponding Author: Dr Bryan Conway

Version 0:

Reviewer comments:

Reviewer #1

(Remarks to the Author)

In their manuscript, Reck et al. conduct a comprehensive multiomic analysis of human kidney disease, specifically identifying an inflammatory tubular cell phenotype. By integrating single-nucleus RNA sequencing and ATAC sequencing with single-cell molecular imaging, the study offers insights into the molecular mechanisms underlying kidney disease pathogenesis. It highlights significant changes of potential key transcription factors, including the loss of HNF4A and activation of AP-1. The research also shows that the inflammatory tubular cells engage a senescence-associated gene program. Targeting AP-1 or the senescence pathway with small molecule inhibitors has shown potential to reduce inflammation and fibrosis in animal models.

Despite the extensive omics analysis performed, the conclusions are constrained by a small sample size and a lack of robust experimental evidence to support the findings. Furthermore, maladaptive proximal tubular cells, similar to the inflammatory tubular cells identified in this study, have been previously described in diseased human kidneys (Lake, B.B. et al., Nature, 2023), which diminishes the novelty of this study.

Major points

1. As shown in Fig. S2K, the correlation within the control group and UO group samples is poor. Therefore, it is necessary to evaluate the variability among samples from different donors within each group to ensure they can adequately represent healthy and diseased states respectively.
2. As shown in Fig. 2C and Fig. S7C, most chemokines are expressed at low proportions in the PT inflammatory cell cluster, and given that these cells originate from different donors, inflammation does not appear to be a predominant feature of this group. Please show the expression of key inflammatory genes across tubular single cells from different donor sources and provide additional evidence to substantiate their inflammatory nature.
3. A similar cell state, named maladaptive proximal tubular cells, has been reported in human diseased kidneys in a previous study (Lake, B.B. et al., Nature, 2023). These maladaptive proximal tubular cells show increased expression of aging-related genes, enhanced activities of stress-response transcription factors, and a senescence-associated secretory phenotype, characteristics that resemble those of the inflammatory tubular cells described in this study. Therefore, a direct comparison of these two cell populations is necessary to elucidate their differences and similarities. Discussion of this point is also warranted.
4. Please explain why different kidney disease samples were used in the CosMx experiment compared to those used in the snRNA-seq experiment. What similarities and differences exist in the molecular phenotypes between the two types of disease samples? Is the inflammatory phenotype of the tubular epithelial cells consistent across these samples?
5. Please validate the co-localization relationships of inflammatory tubular epithelial cells, leukocytes, and myofibroblasts at the protein level using immunofluorescence staining. Also, check the co-localization features in healthy samples.
6. Please validate the role of HNF4A and AP-1 in induction of the inflammatory tubular phenotype by perturbing their

expression in cellular or animal models.

Minor points

1. The heatmap in Fig. 2C and Fig. S6F is duplicated.
2. The data citations from lines 326 to 333 are incorrect; Fig. 6E and Fig. 6F should be referenced as Fig. 7E and Fig. 7F, respectively.
3. In figure 7I, the representative genes list seems to be cherry-picked. It is inconsistent with genes previously listed in fi

(Remarks on code availability)

Reviewer #2

(Remarks to the Author)

Reck and Baird et al. generated multiomics single nucleus datasets combined with single-cell molecular imaging for human diseased kidneys. Their analysis mainly focused on the inflammatory subtype of tubular epithelial cells, marked by enrichment of pro-inflammatory, pro-fibrotic and senescence markers. Their spatial analysis and ligand-receptor pair inference indicated the inflammatory PT cells send the signals to nearby immune cells and myofibroblasts to promote their fibrotic niche in kidney diseases. Their analysis on snATAC-seq characterized disturbance of HNF4A-driven gene regulatory network and activation of NF- κ B and AP-1 transcription factors in the inflammatory PT subset. Finally, they treated disease-model mice with AP-1 / BCL2 inhibitor to show these pathways as potential therapeutic targets.

The reviewer appreciates their effort to generate the high-quality dataset and their state-of-the-art bioinformatics approach. The manuscript is well-written, and the figures are put together nicely. However, the results of their analysis here are viewed as incremental and confirmatory, since they are in line with what has already been reported in other literatures with single-cell / spatial datasets for various kidney diseases. The role of epithelial maladaptive repair in AKI-to-CKD transition as well as implication of AP-1/NFKB/HNF4A transcription factors have been extensively described by other research groups. Furthermore, more comprehensive single cell / spatial datasets such as KPMP dataset have been already published. The inconsistency of the target diseases between the modalities (ie. obstructive nephropathy in single cell analysis vs IGAN/MCNS/pyelonephritis in spatial data) limits deeper analysis of the disease mechanism, leading to just confirmation of previously published findings around shared mechanism of maladaptive epithelial repair and their contribution to CKD. Moreover, for mouse experiments, the effect of AP-1 inhibitor as well as BCL2 inhibitor on mouse kidney disease models have been already published, and their experimental findings were also confirmatory.

In light of these considerations, the authors should address the following important questions: What is the fundamental value / unique contribution of this dataset compared to the existing published datasets? What is the conceptual advance they could demonstrate through analysis of this high-quality dataset?

If the authors think the value lies in the first application of 10Xmultiome kit or CosMx to human kidney disease as in discussion section, they should demonstrate the conceptual advance enabled by these modalities.

Other comments are below:

Majors

1. The authors generated an atlas including obstruction nephropathy kidneys. Are there any differences in gene expression / chromatin accessibility signature of injured/inflammatory PT between obstruction nephropathy (in this dataset) and other kidney disease (eg. AKI, CKD with other etiologies like DN, in previously published dataset)? Recent evidence suggests unique representation of injury/maladaptive repair PT subtypes in mouse UUO compared IRI (Cell Metab 2022 Dec 6;34(12):1977-1998.e9). Therefore, it would be valuable to identify any specific findings unique to the human obstructive nephropathy. Similarly, are there any disease-specific findings in spatial data?
2. The author wrote that the inflammatory PT cells represent a novel subset (P5; 143-144). However, the gene expression signature of this subtype ("inflammatory PT" in Fig2C) shows expression of already established early injury/repair markers (HAVCR1) and mid-late injury/ failed-repair marker (VCAM1, DCDC2, ITGB8 etc), suggesting this subtype is identical to previously described adaptive PT or failed repair PT. In contrast, there is no specific marker gene expression shown in "injured PT" (Fig. 2C). The author should analyze deeper and describe this subtype more clearly. Are they transitional cell state between normal vs adaptive PT? Given overall fuzziness of lineage / injury marker expression in this subtype, this may just reflect low quality cells. Indeed, "injury PT" subtype is not associated with increase in leukocytes and myofibroblasts (P7, 189-190).
3. They used irradiated, senescent RPTEC as a model of inflammatory PT. However, the reviewer suspects if this is a right in vitro model to study inflammatory PT cells, since irradiation induces senescence by double-strand DNA break and other DNA damages. Can the authors show evidence that inflammatory PT cells are potentially induced by DNA damage response (or similar response to other cellular stress) through their bioinformatic approach?
4. Are there any reasonable justifications for generation of spatial kidney data with different diseases (MCNS/IGAN/infection) from multiomics single nucleus data (UUO), as the reviewer believes spatial analysis on the same UUO samples may be more beneficial for the integrative analysis.

5. Do the deconvolution of bulk RNA-seq data (eg cibersort) in Fig 7I show decrease of injured / inflammatory PT by ABT-263 treatment?

Minor

1. P3 72-73: Vcam1+ cells have been already shown to be relevant to human kidney diseases, since they expand in CKD as the author described in the same paragraph.

2. Some well-known TFs regulating epithelial successful vs maladaptive repair (eg SOX9) do not appear in the top list (Fig.6). Can the author discuss this?

3. The reviewer recommends the code availability section to show how to preprocess the dataset. This can be done through a public repository (eg GitHub), at latest before publication.

4. The reviewer recommends an online tool that allows the wet lab researchers who are unfamiliar with bioinformatics to access this dataset.

(Remarks on code availability)

The code was not available in their material. I recommended code availability section.

Version 1:

Reviewer comments:

Reviewer #1

(Remarks to the Author)

The authors have made a commendable effort to address the reviewers' concerns. They've performed additional experiments, including spatial imaging on the nephrectomy samples and multiplex immunofluorescence, which significantly strengthen the study and address the concerns about novelty and validation. The inclusion of a direct comparison with the Lake et al. (2023) dataset and the clarification of the genotype distance data are particularly helpful. The new immunofluorescence data provides valuable mechanistic insights and identifies ICAM-1 as a potential biomarker. While the revisions substantially improve the manuscript, a few points could be further strengthened.

1. Consistency and comparability across different disease samples: Although the authors included CosMx data, the level of coverage across different disease subtypes—both early vs. late CKD stages and various etiologies—may still be limited. While the authors explained the limitations of each model in their response, the manuscript could be further strengthened by including additional discussion on how to overcome the pathological differences between different kidney disease types and by clearly specifying the scope of generalizable conclusions across different diseases.

2. Depth of functional studies on HNF4A: While the authors have provided support for the conclusion that "disruption of the HNF4A network is a major mechanism accounting for the loss of PT identity." The evidence presented in the manuscript is primarily based on single-cell multiomics, chromatin accessibility, and spatial visualization, which are correlational in nature. If feasible, more experimental evidence focusing on HNF4A knockdown/overexpression could be considered to further support its critical regulatory role in the "inflammatory proximal tubule phenotype."

(Remarks on code availability)

Reviewer #2

(Remarks to the Author)

The authors have addressed my concerns with additional data and analyses. Especially, their new spatial dataset on the obstructed kidney tissue strengthened their research. Just as a minor point, the reviewer recommends citing a couple of relevant previous researches that associated ICAM1 expression on PT with kidney disease progression, to highlight the novelty of their finding.

(Remarks on code availability)

They have shown R scripts to generate their figures nicely.

REVIEWER COMMENTS

Reviewer #1 (Remarks to the Author):

In their manuscript, Reck et al. conduct a comprehensive multiomic analysis of human kidney disease, specifically identifying an inflammatory tubular cell phenotype. By integrating single-nucleus RNA sequencing and ATAC sequencing with single-cell molecular imaging, the study offers insights into the molecular mechanisms underlying kidney disease pathogenesis. It highlights significant changes of potential key transcription factors, including the loss of HNF4A and activation of AP-1. The research also shows that the inflammatory tubular cells engage a senescence-associated gene program. Targeting AP-1 or the senescence pathway with small molecule inhibitors has shown potential to reduce inflammation and fibrosis in animal models.

Despite the extensive omics analysis performed, the conclusions are constrained by a small sample size and a lack of robust experimental evidence to support the findings. Furthermore, maladaptive proximal tubular cells, similar to the inflammatory tubular cells identified in this study, have been previously described in diseased human kidneys (Lake, B.B. et al., Nature, 2023), which diminishes the novelty of this study.

Major points

1. As shown in Fig. S2K, the correlation within the control group and UUO group samples is poor. Therefore, it is necessary to evaluate the variability among samples from different donors within each group to ensure they can adequately represent healthy and diseased states respectively.

We apologise that the description in the text did not make it clear that the data in Fig. S2K demonstrates the genotype distance between cells computationally assigned to individual donors, and not transcriptomic distance. Hence, no correlation between individuals from the same group would be expected. As we constructed libraries from pools of nuclei from individual donors, we used a SNP-based computational strategy to infer the original donor of each nucleus. The purpose of Fig S2K is twofold – to match predicted donors (with initially unknown identity) with physical samples and to confirm donors have been correctly inferred. Hence, we expect that a) the genotype of nuclei from the same patient will be similar across different pools; b) using the pooling pattern devised during sample preparation (Fig. S2I) we can identify each sample. This is what we observe in Fig. S2K – we see a pattern of inferred donors having a low genotype distance across libraries - this can be compared to the pooling pattern (Fig. 2I) to identify donors. For example, the genotype distance is very low between cells assigned to patient UUO2

in each of the libraries that contain cells from that donor - libraries 1, 2 and 4. We have altered the legend to Fig. S2K to try to improve the clarity.

Additionally, in order to confirm that predicted cells are assigned to the correct donor, we have genotyped surplus tissue using a SNP microarray. This can be seen in Fig. 2J, with SNP microarray genotypes showing a low distance to their respective predicted genotypes.

We have evaluated the variability between samples within the control and obstructed kidneys in a number of ways. Key clinical and experimental features are similar across patients within the control and UO groups but are significantly different between the groups. For example, the individual patient data for eGFR and histological parameters provided in Suppl. Fig.1 and Suppl. Table 1 confirm that the controls cluster together and are significantly different from patients with ureteric obstruction. In addition, we have now added individual patient datapoints to the proportion of injured and inflamed PT cells (Fig. 2B), demonstrating that the controls have similar proportions of injured/inflammatory PT cells, which is lower than for the UO samples. We also demonstrate in the UMAP in Suppl.Fig.3C that cells from controls are distributed evenly throughout the clusters of healthy tubular cells, whereas injured and inflammatory tubular cells are comprised largely from obstructed kidneys.

2. As shown in Fig. 2C and Fig. S7C, most chemokines are expressed at low proportions in the PT inflammatory cell cluster, and given that these cells originate from different donors, inflammation does not appear to be a predominant feature of this group. Please show the expression of key inflammatory genes across tubular single cells from different donor sources and provide additional evidence to substantiate their inflammatory nature.

It is recognised that chemokines are challenging to detect by single-cell technologies as, due their low mRNA abundance, there is drop out of individual genes at a single-cell level. Hence, the proportion of cells that express the transcript is typically underestimated in snRNA-seq analysis.

For example, the dotplot below demonstrates that while the proportion of inflammatory PT cells that express chemokines is low, it is higher than cells classically considered to be inflammatory such as myeloid and T-cells. The inflammatory PT cells have a unique chemokine expression pattern (CCL20, CCL28, CXCL1, CXCL2, CXCL3, CXCL6), with other cell types expressing higher levels of specific chemokines, for example, macrophages express high levels of CCL2 and CCL8, whereas CCL5 is expressed by T cells.

To determine the relative contribution of inflammatory tubular cells to total chemokine expression in the fibrotic niche, we examined the total number of transcripts (number of transcripts/cell x total number of cells of that type) for each individual cell type. In new **Fig.5E** we show that the inflammatory PT cells account for the majority of cytokine transcripts in the fibrotic niche. Although CCL2 expression is higher in individual macrophages than in inflammatory PT cells, the majority of CCL2 transcripts in the fibrotic niche are derived from inflammatory PT cells due to their higher abundance. The close proximity of chemokine expression in inflammatory PT cells and receptors in adjacent myeloid cells/myofibroblasts facilitates cell-cell signalling and supports a functional effect.

As requested, we provide expression data for key inflammatory genes from different donor sources in the dotplot below. Chemokine expression is consistent across individuals and found exclusively in inflammatory PT cell states (see figure below). This is consistent with the gene expression pattern we found across different technologies (Fig. 2j, Fig. 4f) as well as in the equivalent cells in the KPMP dataset (Lake et al. Fig. S12c). Hence, this provides confidence that these chemokines are expressed specifically by inflammatory PT cells.

3. A similar cell state, named maladaptive proximal tubular cells, has been reported in human diseased kidneys in a previous study (Lake, B.B. et al., Nature, 2023). These maladaptive proximal tubular cells show increased expression of aging-related genes, enhanced activities of stress-response transcription factors, and a senescence-associated secretory phenotype, characteristics that resemble those of the inflammatory tubular cells described in this study. Therefore, a direct comparison of these two cell populations is necessary to elucidate their differences and similarities. Discussion of this point is also warranted.

To provide a clearer assessment of how our inflammatory PT cells related to the adaptive PT cells in the Kidney Precision Medicine Project, we have projected both the injured and inflammatory PT cell transcriptome from our dataset onto the KPMP Atlas dataset (new Fig.2D). While transcripts consistent with injured PT cells, such as *HAVCR1/VCAM1*, are broadly expressed by all adaptive PT cells, the inflammatory PT cells map to a discrete region of the adaptive PT cell cluster, and this region characterised by high expression of cytokines. Hence, pro-inflammatory PTs are a discrete, previously unrecognised subset of the adaptive cell state described in KPMP with a key role in renal inflammation and fibrosis. To confirm inflammatory PT cells are functionally distinct we employed our spatial analysis (Fig 2-5) to demonstrate that it is inflammatory cell state, rather than the injured PT cells, that are co-located in proximity to fibroblasts and immune cells, thus supporting a role for this specific subset in the disease pathogenesis.

4. Please explain why different kidney disease samples were used in the CosMx experiment compared to those used in the snRNA-seq experiment. What similarities and differences exist in the molecular phenotypes between the two types of disease samples? Is the inflammatory phenotype of the tubular epithelial cells consistent across these samples?

We reviewer raises an excellent point, therefore for our revised manuscript we performed high-plex spatial molecular single-cell imaging on the nephrectomy specimens used for the snRNAseq-ATAC analysis, employing ~6,000 gene probes on the CosMx platform (Fig.1F, 2K, 3 and 5).

This new analysis confirmed the findings of our snRNAseq-ATAC analysis, as we could identify distinct injured and inflammatory PT cell states in obstructed kidneys. In addition, our findings replicated the spatial context we observed previously in IgA nephropathy, with specifically the inflammatory PT cells being co-located with immune cells and myofibroblasts in the fibrotic niche. This suggests that this is a stereotypical response to kidney injury and hence therapies that target these cell phenotypes such as AP-1 inhibitors or senolytics may be efficacious across a range of kidney diseases.

Unfortunately, due to the technical differences between our datasets as well as the KPMP cell atlas we were not able robustly perform a direct comparison between disease subtypes. However, we show that the same gene signature is activated in different diseases (e.g. *HAVCR1*, *VCAM1* marks injured PT cells, while *ICAM1*, *CCL2*, *CXCL1* marks inflammatory PT cells).

5. Please validate the co-localization relationships of inflammatory tubular epithelial cells, leukocytes, and myofibroblasts at the protein level using immunofluorescence staining. Also, check the co-localization features in healthy samples.

We have employed multiplex immunofluorescence to identify injured and inflamed proximal tubular (PT) cells at the protein level. Based on the differentially expressed genes from our snRNA-seq analysis (Fig.2C), we have designated injured PT cells as VCAM1+ICAM1- and inflammatory PT cells as VCAM1+ICAM1+ (Fig. 3). Injured and inflamed PT cell phenotypes cluster within individual nephrons as observed previously in the CosMx analysis. Importantly, in agreement with spatial transcriptomics analysis, it was specifically the VCAM1+ICAM1+ inflammatory PT cells that were preferentially surrounded by CD68+ myeloid cells and Fibroblast Activated Protein (FAP)+ myofibroblasts. ICAM1 localised to the luminal membrane adjacent to CD68+ myeloid cells, compatible with a role in disease pathogenesis through tethering immune cells in the urinary space by binding to myeloid receptors including CD11b (*ITGAM*) and CD11c (*ITGAX*). Importantly, ICAM1 may be a clinically useful biomarker to quantify the burden of inflammatory PT cells in kidney biopsies, obviating the need to perform expensive spatial transcriptomics.

6. Please validate the role of HNF4A and AP-1 in induction of the inflammatory tubular phenotype by perturbing their expression in cellular or animal models.

We hypothesise that HNF4A is important in maintaining a healthy proximal tubular cell phenotype rather than in promoting an inflammatory phenotype. Genetic targeting of AP-1 is challenging as the complex constitutes several transcription factors. Therefore, to provide additional support for the role of AP-1 in inhibiting the development of the inflammatory PT cell phenotype, in our murine model we have determined the proportion of PT cells that express both KIM-1 and VCAM1 (unlike in humans, VCAM1 is a specific marker of the inflammatory tubular cell phenotype in mice). We confirmed that the proportion of cells that co-express KIM-1 and VCAM1 cells increases following ischaemia-reperfusion injury, but is reduced following administration of the AP-1 inhibitor (New **Fig.8F**).

Minor points

1. The heatmap in Fig. 2C and Fig. S6F is duplicated.

Apologies again if this was not clear, but the data in Fig. 2C shows the gene expression in nuclei from proximal tubular cell clusters in our nephrectomy specimens, whereas the data in Fig. S6F shows the expression of these same genes in analogous PT cell subsets in the KPMP dataset.

2. The data citations from lines 326 to 333 are incorrect; Fig. 6E and Fig. 6F should be referenced as Fig. 7E and Fig. 7F, respectively.

This has been corrected

3. In figure 7I, the representative genes list seems to be cherry-picked. It is inconsistent with genes previously listed in fi

The datasets are derived from snRNA-seq in PT cells v bulk RNA-seq from whole kidney and human v mouse, therefore it is difficult to compare all genes across both datasets. To avoid any bias from cherry-picking of selective genes, we have examined the entire transcriptome through deconvolution of the bulk RNA-seq as suggested by reviewer 2. This determined that there is a reduction in the proportion of inflammatory PT cells following treatment with ABT263 compared with vehicle (Fig.8I).

Reviewer #2 (Remarks to the Author):

Reck and Baird et al. generated multiomics single nucleus datasets combined with single-cell molecular imaging for human diseased kidneys. Their analysis mainly focused on the inflammatory subtype of tubular epithelial cells, marked by enrichment of pro-inflammatory, pro-fibrotic and senescence markers. Their spatial analysis and ligand-receptor pair inference indicated the inflammatory PT cells send the signals to nearby immune cells and myofibroblasts to promote their fibrotic niche in kidney diseases. Their analysis on snATAC-seq characterized disturbance of HNF4A-driven gene regulatory network and activation of NF- κ B and AP-1 transcription factors in the inflammatory PT subset. Finally, they treated disease-model mice with AP-1 / BCL2 inhibitor to show these pathways as potential therapeutic targets.

The reviewer appreciates their effort to generate the high-quality dataset and their state-

of-the-art bioinformatics approach. The manuscript is well-written, and the figures are put together nicely. However, the results of their analysis here are viewed as incremental and confirmatory, since they are in line with what has already been reported in other literatures with single-cell / spatial datasets for various kidney diseases. The role of epithelial maladaptive repair in AKI-to-CKD transition as well as implication of AP-1/NFKB/HNF4A transcription factors have been extensively described by other research groups. Furthermore, more comprehensive single cell / spatial datasets such as KPMP dataset have been already published. The inconsistency of the target diseases between the modalities (ie. obstructive nephropathy in single cell analysis vs IGAN/MCNS/pyelonephritis in spatial data) limits deeper analysis of the disease mechanism, leading to just confirmation of previously published findings around shared mechanism of maladaptive epithelial repair and their contribution to CKD. Moreover, for mouse experiments, the effect of AP-1 inhibitor as well as BCL2 inhibitor on mouse kidney disease models have been already published, and their experimental findings were also confirmatory.

In light of these considerations, the authors should address the following important questions: What is the fundamental value / unique contribution of this dataset compared to the existing published datasets? What is the conceptual advance they could demonstrate through analysis of this high-quality dataset?

If the authors think the value lies in the first application of 10Xmultiome kit or CosMx to human kidney disease as in discussion section, they should demonstrate the conceptual advance enabled by these modalities.

We thank the reviewer for acknowledging our efforts to create a high quality human multiomic dataset and our use of state-of-the-art bioinformatics to generate high quality figures and well written manuscript.

We have performed additional experiments and re-drafted our manuscript to highlight the key novel insight of our analysis - the identification of highly pathogenic inflammatory PT cells, which are a specific subset of the adaptive cells observed in the Kidney Precision Medicine Project dataset. To clarify this, we include a new Fig.2D, which demonstrates that the inflammatory PT cells map to a discrete region of the adaptive PT cells in the KPMP, which is characterised by high expression of chemokines (Fig.2E). For the first time, we have been able to localise the inflammatory tubular cells by employing high-plex single-cell molecular imaging on the CosMx platform using both a 980-probe panel on human kidney biopsy tissue and in our revised version a 6,000 probe panel on the obstructed kidney tissue employed for the multiome analysis (Fig.2K,3,4,5). This confirms that it is specifically our inflammatory PT cell subset, rather than the injured PT cells, that are co-located with fibroblasts and immune cells, thus supporting a role for this specific subset in the pathogenesis of renal inflammation and fibrosis.

Additionally, we have confirmed these important findings at the protein level by providing additional multiplex immunofluorescence analysis. While *Vcam1* is specific for 'failed repair' or maladaptive PT cells in mice, our snRNA-seq data determined that in humans *VCAM1* is expressed by both injured and inflammatory PT cells, whereas *ICAM1* is a specific marker of the inflammatory PT cell phenotype. Our multiplex immunofluorescence confirmed that specifically the *VCAM1*⁺*ICAM1*⁺ inflammatory tubular PT cells and not the *VCAM1*⁺*ICAM1*⁻ injured tubular cells localise to the inflammatory-fibrotic niche (Fig.3E-L). Hence, an additional finding of our revised paper is that ICAM-1 may be a clinically useful biomarker to quantify the burden of inflammatory PT cells in kidney biopsies, obviating the need to perform expensive spatial transcriptomics.

In addition, the immunofluorescence analysis has provided novel mechanistic insight. We have determined that *ICAM1* is specifically located on the luminal membrane of the inflammatory PT cells and that it acts to tether CD68⁺ myeloid cells to promote further inflammation (Fig.3E), likely through binding to myeloid cell receptors such as CD11b (ITGAM) and CD11c (ITGAX) as determined by our ligand-receptor analysis (Fig.5A) While the AP-1 inhibitor has been previously employed in murine models of sepsis, to our knowledge, our study is the first to determine that AP-1 inhibition ameliorates AKI to CKD transition. We consider that this functional insight is a key strength compare with the manuscript from Lake et al. which, while providing an authoritative multiomic analysis of human kidney disease, is largely descriptive in nature.

Hence, we consider that our findings are novel, rather than confirmatory, as we have identified a novel subset of the adaptive PT cells from the KPMP dataset, identified pro-inflammatory and pro-fibrotic pathways discrete to this subset that may mediate disease and that it is specifically this inflammatory PT cell subset that maps to the inflammatory-fibrotic niche. We propose the use of multiplex immunofluorescence analysis of human biopsy tissue to identify patients who have high proportions of *VCAM1*⁺*ICAM1*⁺ inflammatory PT cells in their kidney biopsies who may be prioritised for therapies such as AP-1 inhibitors or senolytic therapies.

Other comments are below:

Majors

1. The authors generated an atlas including obstruction nephropathy kidneys. Are there any differences in gene expression / chromatin accessibility signature of injured/inflammatory PT between obstruction nephropathy (in this dataset) and other kidney disease (eg. AKI, CKD with other etiologies like DN, in previously published dataset)? Recent evidence suggests unique representation of injury/maladaptive repair PT subtypes in mouse UUO compared IRI (Cell Metab 2022 Dec 6;34(12):1977-1998.e9).

Therefore, it would be valuable to identify any specific findings unique to the human obstructive nephropathy. Similarly, are there any disease-specific findings in spatial data?

We have compared the gene expression signature of our injured/inflammatory tubular cells in the snRNA/ATAC-seq analysis with those from the AKI/CKD datasets. Strikingly, we observe a conserved injury/inflammatory signature regardless of the disease aetiology. Furthermore, we detect these phenotypes in our CosMx dataset from patients with obstructive nephropathy, pyelonephritis and IgA nephropathy. Hence, we consider that these phenotypes represent stereotypical responses to kidney injury, regardless of the disease aetiology and hence our findings are likely to be applicable to a wide range of kidney diseases.

As we have discussed in reviewer 1 (question 4), a direct comparison between disease aetiologies is challenging due to the different technologies used to generate the data. However, we did not see substantive differences in the genes upregulated by both injured and inflammatory PT phenotypes between the disease aetiologies. We hypothesise that it is possible that the differences in the murine models reflect different kinetics of injury rather than disease aetiologies. For example, one tubular cell phenotype was observed predominantly in hyperacute injury (within 6hrs) in the murine IRI model while the other was observed during sustained injury during UUO. As we have employed nephrectomy and renal biopsy tissue from patients with established disease, it is challenging to see changes related to injury kinetics in our dataset, but this will be a focus for future studies.

2. The author wrote that the inflammatory PT cells represent a novel subset (P5; 143-144). However, the gene expression signature of this subtype ("inflammatory PT" in Fig2C) shows expression of already established early injury/repair markers (HAVCR1) and mid-late injury/ failed-repair marker (VCAM1, DCDC2, ITGB8 etc), suggesting this subtype is identical to previously described adaptive PT or failed repair PT. In contrast, there is no specific marker gene expression shown in "injured PT" (Fig. 2C). The author should analyze deeper and describe this subtype more clearly. Are they transitional cell state between normal vs adaptive PT? Given overall fuzziness of lineage / injury marker expression in this subtype, this may just reflect low quality cells. Indeed, "injury PT" subtype is not associated with increase in leukocytes and myofibroblasts (P7, 189-190).

As indicated above, the inflammatory cells are not identical to the adaptive PT cells in the KPMP Atlas, but rather represent a discrete subset (Fig.2D). While they express core injury genes common to the adaptive PT cells in KPMP, such as HAVCR1, VCAM1, DCDC2, ITGB8, they additionally express a unique pro-inflammatory, pro-fibrotic profile (Fig.2C).

We agree with the reviewer that the HAVCR1+VCAM1+ injured PT cells are indeed an intermediate phenotype between healthy and inflammatory PT cells. We suggest that they are not associated with an increase in leucocytes/myofibroblasts precisely because they do not up-regulate pro-fibrotic/pro-inflammatory cytokines until they further differentiate to acquire the inflammatory phenotype. Indeed, our analysis suggests that injured PT cells that are early in the trajectory may have the potential to revert to healthy cells, therefore strategies to deplete all injured PT cells may in fact be deleterious. We do not consider that the injured tubular cells are low quality cells as we employed robust quality control measures to exclude low quality cells. Indeed, our multiplex immunofluorescence confirms at the protein level the presence of discrete phenotypes of VCAM-1⁺/ICAM-1⁻ injured PT cells and VCAM-1⁺/ICAM-1⁺ inflammatory PT cells, with only the latter localising to the fibrotic niche.

3. They used irradiated, senescent RPTEC as a model of inflammatory PT. However, the reviewer suspects if this is a right in vitro model to study inflammatory PT cells, since irradiation induces senescence by double-strand DNA break and other DNA damages. Can the authors show evidence that inflammatory PT cells are potentially induced by DNA damage response (or similar response to other cellular stress) through their bioinformatic approach?

We employed irradiation as an prototypical method of inducing senescence. While irradiation is not the mode of injury in most cases of CKD, it is possible that DNA injury may contribute to the senescent phenotype in CKD patients. We utilised a gamma-H2AX antibody (stains positive nuclei brown in the images below) as a marker of DNA repair following double-strand breaks, to confirm that a higher proportion of tubular cells in the obstructed kidney do exhibit DNA damage, perhaps relating to their senescent phenotype.

4. Are there any reasonable justifications for generation of spatial kidney data with different diseases (MCNS/IGAN/infection) from multiomics single nucleus data (UUO), as

the reviewer believes spatial analysis on the same UUO samples may be more beneficial for the integrative analysis.

As suggested, we have now performed high-plex spatial molecular single-cell imaging on the nephrectomy specimens used for the snRNAseq-ATAC analysis using an increased panel of ~6,000 gene probes on the CosMx platform (Fig 1F, 2K 3 and 5). The additional analysis confirms the presence of injured and inflammatory PT cell subsets as observed in the snRNA-seq of obstructed kidney and that specifically the inflammatory PT cells localise to the fibrotic niche as observed in the spatial analysis of IgA biopsies,

5. Do the deconvolution of bulk RNA-seq data (eg cibersort) in Fig 7I show decrease of injured / inflammatory PT by ABT-263 treatment?

We have performed bulk deconvolution of the bulk RNA-seq as suggested and this does indeed show a decrease in the proportion of cells assigned to an inflammatory phenotype at D42 post-UUO in mice treated with ABT-263 compared with placebo. We have included this in revised Fig.8I

Minor

1. P3 72-73: Vcam1+ cells have been already shown to be relevant to human kidney diseases, since they expand in CKD as the author described in the same paragraph.

We have altered our introduction to highlight that our key novel finding is the identification of a novel subset of the VCAM1⁺ adaptive PT cells that can be identified by expression of ICAM-1 and which adopts a pro-inflammatory, pro-fibrotic signature and localises to the fibrotic niche.

2. Some well-known TFs regulating epithelial successful vs maladaptive repair (eg SOX9) do not appear in the top list (Fig.6). Can the author discuss this?

We do indeed find an enrichment of the SOX9 TF motif (MA0077.1) in the open chromatin of both injured and inflammatory PT cells (as below). Similarly, SOX9 mRNA levels are upregulated in inflammatory PT cells, albeit in a very small proportion of cells, and we have now included SOX9 in dotplots in Fig2C,J and Fig 4F.

The gene regulatory network analysis we have used (SCENIC+ python package) aims to connect TFs to regulatory elements and their target genes. This analysis has several technical limitations (as also discussed in the original SCENIC+ publication), such as overlap between TF motifs, inability to reliably connect distal regulatory elements to target genes, or difficulties due to sparsity of data as typically observed in single-cell data. Hence, we acknowledge that this analysis is not exhaustive and connections may be missed due to these factors. SOX9 is expressed at very low levels and we have observed SCENIC+ struggling to find correlations for both lowly expressed TFs and target genes. Further, with our analysis we aimed to rank TFs by 'importance' based on the number of bindings sites within the regulatory chromatin elements upregulated/opened along the trajectory from healthy to inflammatory phenotypes. We hypothesise this is a relevant metric as a TF that binds to a greater number of regulatory elements will likely have a higher regulatory impact. However, some TFs might have a very narrow, specific set of target gene and hence will not be prioritised by our metric. Given that many of our identified TFs have been recognised by in other publications (e.g. PMID: 38347009), we postulate our importance ranking is broadly effective at identifying relevant TFs.

3. The reviewer recommends the code availability section to show how to preprocess the dataset. This can be done through a public repository (eg GitHub), at latest before publication.

A link to this has now been provided

4. The reviewer recommends an online tool that allows the wet lab researchers who are unfamiliar with bioinformatics to access this dataset.

We have made our data available on the Cell x Gene on-line repository
<https://cellxgene.cziscience.com/collections/307ce143-7cc8-4813-99b1-3797834149c9>

Reviewer #2 (Remarks on code availability):

The code was not available in their material. I recommended code availability section.

A link to this has now been provided

Reviewer #1 (Remarks to the Author):

The authors have made a commendable effort to address the reviewers' concerns. They've performed additional experiments, including spatial imaging on the nephrectomy samples and multiplex immunofluorescence, which significantly strengthen the study and address the concerns about novelty and validation. The inclusion of a direct comparison with the Lake et al. (2023) dataset and the clarification of the genotype distance data are particularly helpful. The new immunofluorescence data provides valuable mechanistic insights and identifies ICAM-1 as a potential biomarker. While the revisions substantially improve the manuscript, a few points could be further strengthened.

1. Consistency and comparability across different disease samples: Although the authors included CosMx data, the level of coverage across different disease subtypes—both early vs. late CKD stages and various etiologies—may still be limited. While the authors explained the limitations of each model in their response, the manuscript could be further strengthened by including additional discussion on how to overcome the pathological differences between different kidney disease types and by clearly specifying the scope of generalizable conclusions across different diseases.

- We have expanded our discussion to discuss how the injured and inflammatory tubular cell phenotypes may represent a stereotypical response to injury as they are conserved across multiple kidney diseases (Page 13, second paragraph). We agree that additional studies using standardised methodologies across multiple aetiologies and stages of kidney disease will be required to identify disease and stage-specific cellular phenotypes.

2. Depth of functional studies on HNF4A: While the authors have provided support for the conclusion that "disruption of the HNF4A network is a major mechanism accounting for the loss of PT identity." The evidence presented in the manuscript is primarily based on single-cell multiomics, chromatin accessibility, and spatial visualization, which are correlational in nature. If feasible, more experimental evidence focusing on HNF4A knockdown/overexpression could be considered to further support its critical regulatory role in the "inflammatory proximal tubule phenotype."

- The experiments proposed by the reviewer have previously been conducted in both *in vivo* (29) and *in vitro* (30) studies. Knockout of *Hnf4a* in tubular cells in mice inhibits maturation of the proximal tubule, resulting in Fanconi syndrome (29). In addition, studies employing *HNF4A* knockout or over-expression in human iPS cell-derived kidney organoids have demonstrated a role for HNF4A in promoting a more mature proximal tubular cell phenotype (30). Furthermore, *HNF4A* knockout reduced

expression of key proximal tubular cell genes in organoids, and we have performed additional analysis which demonstrated that these genes overlapped significantly with our predicted HNF4A target genes (new Fig.S20a, additional methods, page 25, second paragraph). The genes that were down-regulated in *HNF4A* knockout versus wild type organoids were also enriched amongst genes that exhibited reduced expression in inflamed versus healthy proximal tubular cells in our snRNA-seq analysis (New Fig.S20b). This provides experimental support for our hypothesis that HNF4A is critical for maintenance of a healthy proximal tubular cell phenotype and that loss of HNF4A may facilitate adoption of the inflammatory phenotype. We have discussed this in our revised manuscript (Page 9, last paragraph, blue font).

Reviewer #2 (Remarks to the Author):

The authors have addressed my concerns with additional data and analyses. Especially, their new spatial dataset on the obstructed kidney tissue strengthened their research. Just as a minor point, the reviewer recommends citing a couple of relevant previous researches that associated ICAM1 expression on PT with kidney disease progression, to highlight the novelty of their finding.

We have referenced prior studies that linked proximal tubular expression of ICAM-1 to kidney disease in the discussion (Page 14, 1st paragraph).

Reviewer #2 (Remarks on code availability):

They have shown R scripts to generate their figures nicely.

We have included the following statement in the discussion 'Further studies using standardized methodologies across a wide spectrum of kidney diseases and stages of disease progression will be required to identify disease- and stage-specific pathogenic cell phenotypes to enable targeted therapies. In particular, biopsies from early disease may be helpful in determining the sequence of events that initiate development of the fibrotic niche'.